# Closed universes in two dimensional gravity

**Mykhaylo Usatyuk[1], Zi-Yue Wang[2], Ying Zhao[1]**

[1] *Kavli Institute for Theoretical Physics, Santa Barbara, CA 93106, USA*

[2] *Department of Physics, University of California, Santa Barbara, CA 93106, USA*

musatyuk@kitp.ucsb.edu, zi-yue@ucsb.edu, zhaoying@kitp.ucsb.edu

### Abstract

We study closed universes in simple models of two dimensional gravity, such as Jackiw-Teiteilboim (JT) gravity coupled to matter, and a toy topological model that captures the key features of the former. We find there is a stark contrast, as well as some connections, between the perturbative and non-perturbative aspects of the theory. We find rich semi-classical physics. However, when non-perturbative effects are included there is a unique closed universe state in each theory. We discuss possible meanings and interpretations of this observation.

# 1  Introduction

Over the past decades we have learned a lot about quantum gravity in asymptotically AdS spacetimes through the AdS/CFT correspondence [1–3]. Quantum gravity in spacetimes without AdS asymptotics is less well understood. The universe we live in may not have an asymptotic boundary, and is more similar to de Sitter due to the positive cosmological constant.

De Sitter space has many mysterious features. It does not have a spatial boundary and it is exponentially expanding. Studying de Sitter space directly is quite challenging, but see [4–9] for some recent works. In this paper we will study the simpler problem of AdS cosmologies, i.e., closed universes with negative cosmological constant. They are FRW cosmologies where the universe begins with a big bang and ends with a big crunch. Like de Sitter space, they also do not have spatial boundaries. Unlike de Sitter space, they do not expand forever in the future. See Figure 2. Our hope is that AdS cosmologies are easier to study than de Sitter space due to our understanding of AdS/CFT, and we may learn some general lessons about closed cosmologies which can be applied to de Sitter space and potentially our universe.

Spacetimes without spatial boundaries are very different from spacetimes with boundaries such as asymptotically anti-de Sitter or asymptotically flat spacetimes. One feature of asymptotically anti-de Sitter space is that it has a fixed, cold boundary where gravity is turned off. It offers a natural location where a dual quantum mechanical system can live. It can also play the role of an observer relative to which we can describe bulk physics and define bulk operators. Without spatial boundaries whether or not a closed universe has a dual non-gravitational description, or what the structure of such a dual is, is unclear.

Closed universes in AdS space have been studied in higher dimensions in a variety of works. A common approach to study Lorentzian AdS cosmologies is to first find a Euclidean wormhole connecting two asymptotically AdS boundaries [10], and analytically continue the geometry to get a big bang/big crunch cosmology without any asymptotic boundaries. Maldacena-Maoz constructed such wormholes with the aim of understanding closed universes in AdS/CFT [10]. The implication of Maldacena-Maoz like wormholes for cosmology was discussed by McInnes in [11, 12], and more recently investigated by [13–19]. These higher dimensional investigations have largely been focused on perturbative questions.

In this paper we will take one step in exploring closed universes without boundaries, both perturbatively and non-perturbatively. We will study simple toy models in two dimensions where calculations are more tractable than higher dimensions. We will see that there is a stark contrast, as well as some connections, between semi-classical and non-perturbative aspects of the theory of closed universes. A very interesting and important question is how to reconcile these two aspects. We will not be able to answer this question in general in this paper. Instead, we will point out various features occurring in simple models and discuss the lessons we learn from them.

**Summary of results.**

**Perturbative construction.** In section 2 we study perturbative aspects of closed universes. The model we consider is JT gravity coupled to matter fields $\mathcal{O}_i$. We will be considering cosmological states of closed universes on Cauchy slices with topology $S^1$. We specify a cosmological state by inputting a set of asymptotic boundary conditions, and performing the gravity path integral up to a compact slice of the closed universe. We consider boundary conditions specified by an insertion $\text{tr}(e^{-\beta H}\mathcal{O}_i)$ where $\mathcal{O}_i$ has vanishing one-point function. We find the dominant saddle point configuration is a trumpet geometry that ends on a compact circle, with a massive particle on the slice, see figure 1. This gives a wave functional for a closed universe on a circle. We can then perform Lorentzian time evolution of the closed universe, where it collapses in a big crunch.

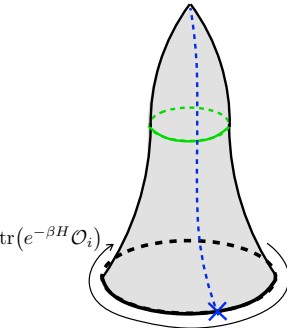

Figure 1: Preparing a closed universe state with a particle (dashed blue).

We can prepare an infinite number of seemingly orthogonal cosmological states by changing the asymptotic boundary conditions. For example, by exciting different flavors $i, j$ of matter fields at the asymptotic boundary

$$|\psi_{\beta,i}\rangle = \quad\raisebox{-1em}{$\underset{\mathcal{O}_i}{\bigcirc}$}\quad , \qquad |\psi_{\tilde{\beta},j}\rangle = \quad\raisebox{-1em}{$\underset{\mathcal{O}_j}{\bigcirc}$}\quad , \qquad \langle\psi_{\beta,i}|\psi_{\tilde{\beta},j}\rangle \propto \delta_{ij} . \tag{1.1}$$

These states clearly appear orthogonal. If we consider the inner product matrix defined by $M_{ij} = \langle i|j\rangle$ then it appears we can make its rank as large as we want. Namely, we can construct semi-classical states with incredibly varied physics. We will see this is no longer true when non-perturbative effects are included.

**Non-perturbative physics.** In section 3 we consider a topological toy model that mimics the key features of JT coupled to matter fields (1.1). This model is motivated by the topological model studied by Marolf and Maxfield in [20]. The model is defined by defining states to live on circles with an insertion carrying a flavor index $i = 1, ..., k$. Observables specified by some set of boundary conditions

are evaluated by summing over all bulk manifolds $\mathcal{M}$ and over all pairwise contractions of matching flavor indices on $\mathcal{M}$:

$$|\psi_i\rangle \equiv \quad \bigcirc_{i} \quad , \qquad \langle\psi_i|\psi_j\rangle = \quad {}_{i}^{i} \Big|\Big| \quad + \ldots , \qquad I(\mathcal{M}) = -S_0\chi + I_{\text{matter}} . \qquad (1.2)$$

The action weighs bulk manifolds by the Euler Characteristic $\chi$, and $I_{\text{matter}}$ enforces that all matter flavors $i$ must pairwise contract, otherwise the manifold $\mathcal{M}$ does not contribute. For example, in the above picture we have $\langle\psi_i|\psi_j\rangle = \delta_{ij} + \mathcal{O}(e^{-2S_0})$. Thus, the inner product matrix $M_{ij} = \langle\psi_i|\psi_j\rangle$ appears to have a large rank due to the large number of seemingly orthogonal states. However, when including the effect of spacetime wormholes, we will show the following:

- Non-perturbative wormhole effects dramatically modify the above inner product. The dimension of the closed universe Hilbert space is reduced to one.

- The topological model computes an ensemble average over distinct theories ($\alpha$-sectors). We study the probability distribution for the ensemble.

In each $\alpha$-sector, the matrix $M$ has rank one with a single eigenvalue given by $\operatorname{tr} M$. We elaborate on this argument shortly. $\operatorname{tr} M$ should be thought of as a random variable with distribution over different $\alpha$-sectors. We will find that in this model it is the product of two independent random variables:

$$\operatorname{tr} M = Z\left(\sum_{i=1}^{k} A_i^2\right). \qquad (1.3)$$

The first factor, $Z$, is the dimension of the Hilbert space of two-sided wormholes in each $\alpha$-sector and obeys Poisson statistics. As a boundary Hilbert space dimension, the eigenvalues of $Z$ are discrete and equally spaced. The second factor, $\sum_{i=1}^{k} A_i^2$, comes from pairing up flavor indices and obeys the chi-squared distribution. It can be thought of as the square of the matter thermal one-point function $\langle\mathcal{O}_i\rangle^2$ in a fixed member of the ensemble. The number $A_i \in \mathbb{R}$ takes continuous values.[1]

More generally, the matrix elements of $M$ take the form

$$M_{ij} = Z\left(A_i A_j\right), \qquad (1.4)$$

as an equality between random variables.

**Non-perturbative Hilbert space.** We give the general argument that the dimension of the Hilbert space of quantum gravity on compact Cauchy surfaces is one dimensional. This argument was given

---

[1] The variables $A_i$ are drawn from a gaussian distribution, with zero mean and variance one.

in [21] in the context of dS and we summarize and generalize it here. The argument relies on defining the inner product between states through the gravitational path integral [22], and is insensitive to the details of the theory of interest and even the dimensionality of the spacetime.[2] The key ingredient is that the states live on slices without boundaries.

The starting point is to define states on compact Cauchy slices. Since we will be general, we will take states $|i\rangle, |j\rangle$ to be labelled by generic boundary conditions on the slice.[3] The inner product will be defined by

$$\langle i|j\rangle = \int \mathcal{D}g \, e^{-I[g]}, \qquad \qquad \qquad \qquad \tag{1.5}$$

In the above we sum over all manifolds $\mathcal{M}$ with boundaries given by Cauchy slices with boundary conditions $i, j$, and over all Euclidean metrics $g$.[4] To confirm that states are orthogonal we should calculate higher moments to check that the variance is small [21, 23]. Doing this for the inner product between seemingly orthogonal states $\langle i|j\rangle = \delta_{ij}$, we find

$$|\langle i|j\rangle|^2 = \langle i|i\rangle\langle j|j\rangle + \dots, \tag{1.6}$$

Since the states live on slices without boundaries, bulk manifolds can cross connect different bras and kets, and give large corrections to inner products. The above fact implies that the dimension of the Hilbert space is greatly reduced. More precisely, consider the Gram matrix $M_{ij} = \langle i|j\rangle$ built from a set of states $|i\rangle$ with $i = 1, \dots, k$. The number of non-perturbatively orthogonal states is given by

$$\text{rank}(M) = \lim_{n \to 0} \text{Tr}\left(M^n\right). \tag{1.7}$$

The right-hand side needs to be evaluated for all integer $n$ using the gravity path integral, and then analytically continued to take the limit. When $\text{Tr}\left(M^n\right)$ is calculated it can immediately be seen that $\text{Tr}\left(M^n\right) = (\text{Tr}\,M)^n$. This is true because the boundary conditions can be permuted without changing the value of the path integral, see section 3. This immediately implies that $\lim_{n \to 0} \text{Tr}\left(M^n\right) = 1$, and we conclude

$$\dim(\mathcal{H}_{\text{closed}}) = 1. \tag{1.8}$$

All states defined on compact Cauchy slices turn out to be equivalent non-perturbatively.

---

[2] One caveat is that the gravity path integral is not as well defined in higher dimensions and with general theories. The argument relies on formal manipulations of higher moments of inner products in higher dimensions, where higher moments may not have convergent answers if taken literally. The conclusion regarding the Hilbert space dimension being one relies on an analytic continuation of these divergent moments to a zeroth power.

[3] For example, in 4d general relativity the boundary conditions label different three metrics, whereas in pure JT the boundary conditions would label the size $b$ of geodesic circles from section 2.

[4] In principle the manifold could be multiple disjoint manifolds.

## Outline

This paper is organized as follows. In section 2 we study perturbative aspects of the theory of closed universes in JT gravity plus matter. We obtain wave functionals on spatial slices and show that there is rich semi-classical physics. In section 3 we include non-perturbative effects in a simple topological model that mimics JT coupled to matter. We show that if we include the effects of spacetime wormholes, there is only one closed universe state in each $\alpha$-sector. We also obtain the distribution of the norm-squared of this state and discuss its physical meaning. In section 4 we point out unanswered questions and future directions.

The goal of this work is to study closed universes in the simplest setting where non-perturbative effects can be included and are under complete control. The aim is to understand the framework for describing physics in closed universes. We will find that various observables and calculations that are well defined perturbatively seem to become ill-defined when non-perturbative effects are included. There have been other inspiring works studying non-perturbative aspects of closed universes. In particular, [10] and [24] contain a lot of comments which overlap with what we discuss.

## 2 Perturbative Aspects of JT closed universes

In this section we study the perturbative physics of a closed universe in JT gravity. We first consider the case without matter as a warm-up, then we study the case with matter. We will see that by specifying different asymptotic boundary conditions, we can construct a large number of orthogonal semi-classical states that describe very different closed universes.

### 2.1 Closed universes in JT

We first consider Lorentzian JT gravity on the cylinder topology $S^1 \times \mathbb{R}$. The Lorentzian action is given by [25][5]

$$I[g, \Phi] = \frac{1}{2} \int d^2x \sqrt{-g} \Phi \left( R + 2 \right) , \tag{2.1}$$

in terms of the dilaton $\Phi$ and metric $g$. The equations of motion are

$$R + 2 = 0, \qquad \left( -\nabla_\mu \nabla_\nu + g_{\mu\nu} \nabla^2 - g_{\mu\nu} \right) \Phi = 0 . \tag{2.2}$$

The classical solutions are given by

$$ds^2 = -dt^2 + b^2 \cos^2(t) d\sigma^2, \qquad \Phi(t) = \phi_c \sin(t) , \tag{2.3}$$

---

[5] Throughout the paper we set $8\pi G_N = 1$.

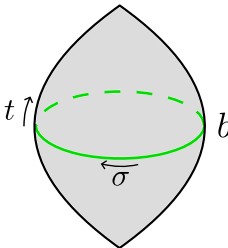

Figure 2: Classical solution for a closed universe. It begins at a big bang and re-collapses in a big crunch. The green geodesic is the maximal slice of the universe.

with compact spatial coordinate $\sigma \sim \sigma + 1$ and time $t \in \left(-\frac{\pi}{2}, \frac{\pi}{2}\right)$. See figure 2. The space of solutions is labelled by $b > 0$ and $\phi_c \in \mathbb{R}$. This geometry corresponds to a big bang/big crunch universe that comes into existence at $t = -\frac{\pi}{2}$ from a conical singularity and expands before re-collapsing into a conical singularity at $t = \frac{\pi}{2}$. The universe reaches a maximal spatial size $b$ at $t = 0$, which is the only closed geodesic on the geometry. The dilaton functions as a clock, attaining maximal amplitude $|\phi_c|$ at the big bang/crunch.

**Canonical quantization.** The classical theory is very simple due to the lack of asymptotic boundaries, and has no apparent interesting dynamics. Phase space is labelled by the maximal size of the universe and the value of the dilaton $(b, \phi_c)$. The quantization of this system is straightforward to carry out (see [26] for the case with boundaries) [27–29]. The symplectic form can be evaluated to find

$$\omega = \delta\phi_c \wedge \delta b, \tag{2.4}$$

so $b$ and $\phi_c$ are conjugate variables. We can canonically quantize in the $b$ basis obtaining a basis of delta function normalized states $|b\rangle$, with $\langle b'|b\rangle = \frac{1}{b}\delta(b - b')$ being the natural inner product from the Euclidean path integral. We can prepare natural wavefunctions for these closed universes by doing Euclidean path integrals over, for example, a trumpet geometry ending on a geodesic of size $b$. Such wavefunctions are dominated by small universes with $b \to 0$ and so do not appear particularly interesting.

Another approach to quantization is through the Wheeler-DeWitt wavefunctional. Leaving technical details to appendix A.1, the WdW wavefunction for 2d dilaton gravity theories on spatial slices $\Sigma = S^1$ was solved in [28, 30, 31]. The input is given by the value of the dilaton $\phi(\sigma)$ and the induced metric $h$ along the cauchy slice. There are two branches of solutions [31]

$$\Psi^{\pm}_{\phi_c^2}[\phi(\sigma), h] = \exp\left(\pm i \int_0^1 d\sigma \sqrt{h}\left[\sqrt{\phi_c^2 - \phi^2 + \phi'^2} - \phi' \tanh^{-1}\left(\sqrt{\frac{\phi_c^2 - \phi^2}{\phi'^2} + 1}\right)\right]\right), \tag{2.5}$$

where there is a continuous family of solutions to the WdW equation labelled by a parameter $\phi_c^2$ which turns out to be dilaton squared. Following Hartle-Hawking [22] the amplitude squared $|\Psi^{\pm}_{\phi_c^2}[\phi(\sigma), h]|^2$

roughly tells us the likelihood to find a Cauchy slice with field configuration $(\phi(\sigma), h)$ in the state with fixed $\phi_c^2$. We restrict to the case of a constant dilaton profile $\phi_0$ on a slice of length $L$. We find that to reproduce semiclassical expectations of the geometry we must carefully choose an $i\epsilon$ prescription to avoid the branch cut

$$\Psi_{\phi_c}[\phi_0, L] = \exp\left(i\phi_c L\sqrt{1 - \frac{\phi_0^2}{\phi_c^2} + i\epsilon\phi_c}\right). \tag{2.6}$$

Note that if we restrict to the geodesic slice where $\phi_0 = 0$, we recover $e^{i\phi_c b}$, which is consistent with the fact that $b$ and $\phi_c$ are conjugate variables. To see that this wavefunction (2.6) reproduces semiclassical expectations it is more intuitive to work in the $|b\rangle$ basis. Since $b$ is canonically conjugate to $\phi_c$ we change basis to find

$$\begin{aligned}
\Psi_b[\phi_0, L] &= \int_{\mathbb{R}<0} d\phi_c e^{-ib\phi_c} e^{-iL\sqrt{\phi_c^2 - \phi_0^2 - i\epsilon}} + \int_{\mathbb{R}>0} d\phi_c e^{-ib\phi_c} e^{iL\sqrt{\phi_c^2 - \phi_0^2 + i\epsilon}}, \\
&\approx \begin{cases} 2\cos\left(\phi_0\sqrt{b^2 - L^2}\right), & L < b, \\ e^{-\phi_0\sqrt{L^2 - b^2}}, & L > b. \end{cases}
\end{aligned} \tag{2.7}$$

We see the result for the amplitude is quite reasonable. In an eigenstate $|b\rangle$ the universe attains a maximal size $b$ and has slices of all smaller lengths $L < b$. The wavefunction oscillates in this regime, signaling classically allowed behavior. In the classically forbidden regime $L > b$ the wavefunction exponentially decays.

## 2.2 Closed universes in JT with matter

We will now consider JT gravity coupled to matter. Pure JT is exactly solvable but lacks a number of key features that give rise to a rich set of perturbative physics in closed universes. Primarily, by including matter we can introduce a notion of an observer living in a closed universe. We will again construct a large class of seemingly orthogonal semiclassical states.

The model we consider is JT gravity coupled to a massive particle.[6] The Lorentzian action is given by[7]

$$I[g, \Phi] = \frac{1}{2}\int d^2x\sqrt{g}\,\Phi\,(R + 2) - m\int ds\sqrt{g_{\mu\nu}\dot{X}^\mu \dot{X}^\nu} + \int_{\text{bdy}}\sqrt{h}\,\Phi_b(K - 1), \tag{2.8}$$

where $X^\mu(s)$ is the worldline of the massive particle with $s$ an affine time along the worldline. The

---

[6] For some discussions when we have massive scalar field, see appendix A.4 .

[7] The boundary term vanishes and does not contribute on the Lorentzian cosmology. We include it for the Euclidean discussion later.

equations of motion are

$$R + 2 = 0, \qquad \left(-\nabla_\mu\nabla_\nu + g_{\mu\nu}\nabla^2 - g_{\mu\nu}\right)\Phi(x^\mu) = m\int ds \frac{\dot{X}_\mu\dot{X}_\nu}{\sqrt{\dot{X}^2}}\delta^{(2)}\left(X^\mu(s) - x^\mu\right),\qquad (2.9)$$

with the particle following a geodesic. The stress tensor of the particle implies a jump in the dilaton $n^\mu\partial_\mu\Phi = m$ across the particle, where $n$ is the normal vector to the particle. The classical solutions are

$$ds^2 = -dt^2 + b^2\cos^2(t)d\sigma^2, \qquad \Phi(t) = \frac{m}{2\sinh\frac{b}{2}}\cos(t)\cosh(b\sigma) + \phi_c\sin(t),\qquad (2.10)$$

with the particle sitting for at $\sigma = \pm\frac{1}{2}$ for all time which is identified under $\sigma \sim \sigma + 1$.[8] The first term in the dilaton has discontinuous derivative across the particle. Note that again the classical solutions are characterized by two parameters, the size of the universe $b$ and the dilaton value $\phi_c$. The symplectic form can again be evaluated and found to be $\omega = \delta\phi_c \wedge \delta b$ indicating that $\phi_c$ and $b$ are conjugate variables, just as in the case of pure JT. We can again quantize in the $|b\rangle$ basis and obtain a set of states of closed universes with a massive particle on the Cauchy slice.

### 2.2.1 Euclidean path integral construction

We would like to construct a large basis of closed universe states by using Euclidean path integral techniques. Following [22] we can specify a closed universe state by specifying boundary conditions and performing a path integral up to a compact slice. With the no-boundary proposal in [22], the wave functional $\Psi_{\text{NB}}[b]$ is obtained by integrating over all Euclidean geometries ending on a circle of size $b$ and no other boundaries.

In our context we will specify cosmological states by choosing a different set of boundary conditions. We specify our boundary condition by the following quantity: $\text{tr}\left(e^{-\beta H}\mathcal{O}_i\right)$, where $H$ should be thought of as the boundary Hamiltonian dual to JT + matter and $\mathcal{O}_i$ is a matter operator insertion [32]. This maps to a Euclidean boundary condition given by a circular boundary of length $\beta$ with an operator insertion $\mathcal{O}_i$. We will treat Euclidean wormholes between these boundaries as computing an inner product between different cosmological states. We can then cut apart the Euclidean wormhole along a geodesic slice $b$ and analytically continue the geometry to get a closed universe.

In JT with matter the Euclidean wormhole that will be important for us is the double trumpet geometry. The metric is given by

$$ds^2 = d\rho^2 + b^2\cosh^2(\rho)d\sigma^2,\qquad (2.11)$$

where $\rho \in (-\infty, \infty)$ and again $\sigma \sim \sigma + 1$. The two asymptotic boundaries are at $\rho \to \pm\infty$, and the only closed geodesic is at $\rho = 0$. By analytically continuing along the time symmetric slice $\rho \to it$ we

---

[8] With the particle present, we must explicitly restrict to $\sigma \in [-\frac{1}{2}, \frac{1}{2}]$.

obtain the closed universe cosmology (2.10) [10]. See Figure 3.

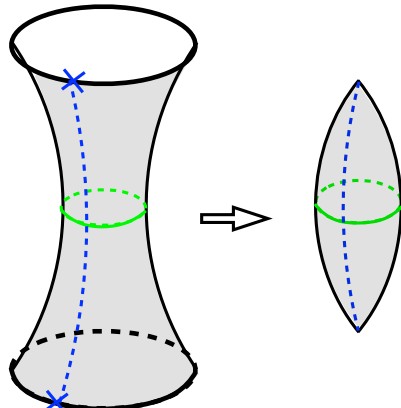

Figure 3: Matter stabilized Euclidean wormhole (left), and the analytic continuation (right) to real time. The wormhole geometry becomes a closed universe with a heavy massive particle (dashed blue), with a maximal geodesic slice (green circle).

**Comparison with black hole case.** It is instructive to contrast our approach with the standard way of constructing the thermofield double state (TFD). In preparing the TFD one specifies Euclidean time evolution $e^{-\frac{\beta H}{2}}$ on the half circle. With this boundary condition we get a wave-functional on spatial slices, and the leading saddle is eternal black hole with temperature $\frac{1}{\beta}$, see figure 4(a).

For the closed universe the boundary condition is a full circle represented by $\text{tr}\big(e^{-\beta H}\mathcal{O}_i\big)$. We choose $\mathcal{O}_i$ such that its one-point function vanishes, which implies that the disk amplitude vanishes. As a result we get a wave-functional on compact slices, with the leading saddle corresponding to closed universe of size $b$, see figure 4(b). The double trumpet geometry computes the norm squared of this state $\big[\text{tr}\big(e^{-\beta H}\mathcal{O}_i\big)\big]^2$.

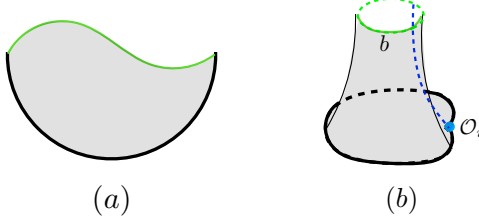

$(a)$        $(b)$

Figure 4: Boundary conditions preparing a black hole state and a closed universe state

**Closed universe with one observer**

The classical wormhole with a massive particle can be found by finding the saddlepoint for the quantity $\big(\text{Tr}[e^{-\beta H}\mathcal{O}_i]\big)^2$. This amounts to a geometry problem where we must cut out a double trumpet from the hyperbolic disk with a worldline running through the center, which is a standard construction [33–35].

In appendix A.2 we give details on constructing the solution. The final result is there is a classical wormhole with geodesic throat size

$$b_0 \approx 2 \log\left(\frac{m\beta}{\pi\phi_r}\right), \tag{2.12}$$

In the above we have taken the limit of $m \gg \frac{\phi_r}{\beta} \gg 1$ for a large wormhole. With matter, the dilaton has an on-shell solution given by

$$ds^2 = d\rho^2 + b_0^2 \cosh^2(\rho)d\sigma^2, \qquad \Phi(\rho,\sigma) \approx \frac{\pi\phi_r}{\beta}\cosh(\rho)\cosh(b_0\sigma), \tag{2.13}$$

where we have expanded the overall prefactor of the dilaton at large $b_0$. Taking this solution and analytically continuing $\rho \to it$ along the time symmetric slice we arrive at a large semiclassical closed universe with a massive particle

$$ds^2 = -dt^2 + b_0^2 \cos^2(t)d\sigma^2, \qquad \Phi(t,\sigma) \approx \frac{\pi\phi_r}{\beta}\cos(t)\cosh(b_0\sigma). \tag{2.14}$$

We show the construction along with the analytic continuation in Figure 5.

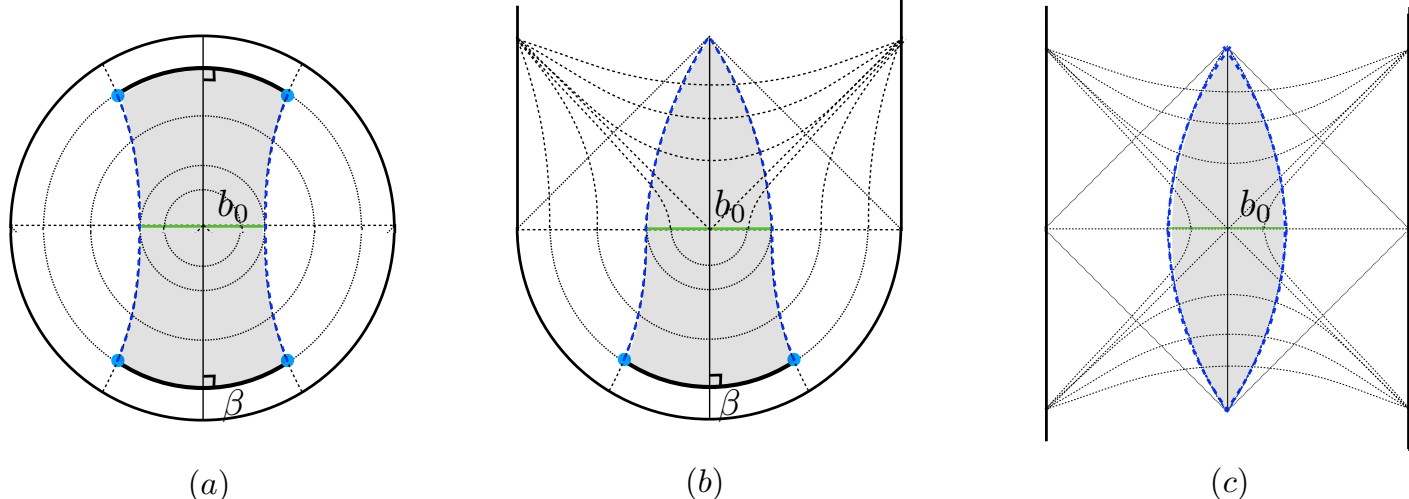

$(a)$ $\qquad\qquad\qquad\qquad (b)$ $\qquad\qquad\qquad\qquad (c)$

Figure 5: The grey lines are lines of constant dilaton. (a) shows the double trumpet solution embedded into the hyperbolic disk, with the blue line the massive particle. The green line is the geodesic throat $b_0$. The two blue lines are geodesics and are identified. (b) shows the analytic continuation of the double trumpet into Lorentzian time $\rho \to it$, where it becomes a closed universe. (c) shows the closed universe solution embedded into a WDW patch.

**Adding additional particles**

We can also solve for the classical geometry perturbatively with additional particles. Assume that in addition to the observer with mass $m_1$, we also have a light particle with mass $m_2$. We consider the quantity $\left(\text{Tr}[\mathcal{O}_1 e^{-\beta_1 H}\mathcal{O}_2 e^{-\beta_2 H}]\right)^2$ where $\beta_1 + \beta_2 = \beta$ and assume $m_1 \gg \frac{\pi\phi_r}{\beta} \gg m_2$, The Euclidean

double trumpet geometry, along with the dilaton profile, can be solved using hyperbolic geometry and charge conservation which we do in appendix A.3. The geometry can then be analytically continued to obtain a closed universe with two matter excitations.

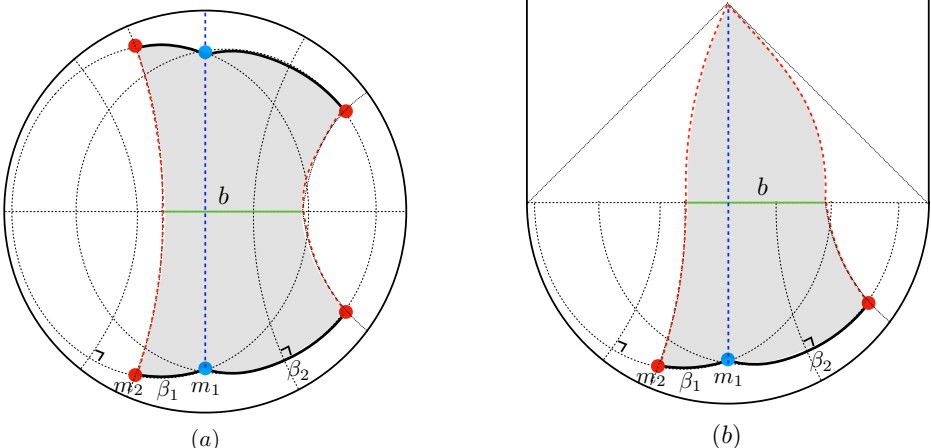

$(a)$ $\qquad\qquad\qquad\qquad\qquad$ $(b)$

Figure 6: (a) The euclidean double trumpet with two matter excitations. (b)Analytic continuation of (a) from time-reflection symmetric slice.

The geometry is denoted in Figure 6. The two red lines are identified and represents the light particle $m_2$. The blue line represents the heavy observer $m_1$. The size of the closed universe $b$ is given by

$$b = b_0 + \frac{m_2\beta}{\pi\phi_r}\left[\tan\left(\frac{\pi\beta_1}{2\beta}\right) - \frac{2}{\pi}\left(1 - \frac{\frac{\pi\beta_1}{\beta}}{\tan\left(\frac{\pi\beta_1}{\beta}\right)}\right)\right] + \mathcal{O}\left((m_2\beta/\phi_r)^2\right) \tag{2.15}$$

where $b_0$ is the size of the universe with only the heavy observer (2.12). One feature of equation 2.15 is that as we move the light particle relative to the heavy particle, with $\beta$ fixed, the size of the universe $b$ will change. The dilaton is slightly more complicated, and given in equation (A.23).

### 2.2.2 Wave functional of a closed universe with one observer

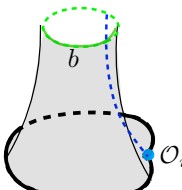

Figure 7: Given boundary conditions $\mathrm{tr}\left(e^{-\beta H}\mathcal{O}_i\right)$, we compute the wave functional in the $|b\rangle$ basis by integrating over trumpet geometries.

After obtaining classical solutions, we now look for wave functionals on spatial geometries. Given boundary condition $\mathrm{tr}\!\left(e^{-\beta H}\mathcal{O}_i\right)$, we will compute the wave functional in the $|b\rangle$ basis by integrating over trumpet geometries (Figure 7). The Euclidean action is given by

$$I = -\frac{1}{2}\int \sqrt{g}\,\Phi(R+2) - \int_{\mathrm{bdy}} \sqrt{h}\,\Phi_b(K-1) + mL\,. \tag{2.16}$$

We perform the computation where only the dilaton is allowed to be off-shell.[9] Details are left to appendix A.4, with the final result for the wavefunctional of the universe created by $\mathrm{tr}\!\left(e^{-\beta H}\mathcal{O}_i\right)$ in $|b\rangle$ basis given by

$$\Psi_\beta(b) = \exp\left(-m\log\left(\frac{\beta/\phi_r}{\cosh\left(\frac{b}{2}\right)\arctan\left(\sinh\frac{b}{2}\right)}\right) + \frac{2\phi_r}{\beta}\arctan^2\left(\sinh\frac{b}{2}\right) - \frac{4\phi_r}{\beta}\sinh\left(\frac{b}{2}\right)\arctan\left(\sinh\frac{b}{2}\right)\right).$$
$$\tag{2.17}$$

It can be checked that the saddlepoint for this wave functional at large $m$ is at $b_0 = 2\log\left(\frac{m\beta}{\pi\phi_r}\right)$, which corresponds to the classical solution where all modes are on-shell. Here we ignore Schwarzian and matter fluctuations by taking $\beta/\phi_r \ll 1$ and $m \gg 1$. The amplitude is highly peaked around the classical value.

Similar to our calculation of the WdW wavefunctional $\psi_b[\phi_0, L]$ in the case without matter, we can repeat the procedure with matter for a generic slice of length $L$ with dilaton profile $\Phi(\sigma)$. We do this in appendix A.5 for a profile $\Phi(\sigma) = B\cosh(b\sigma)$. The final wavefunctional is given by

$$\Psi_\beta[L,\phi] = \int db\,\Psi_\beta(b)\psi_b(L,\Phi)\,, \tag{2.18}$$

where $\psi_b(L,\Phi)$ is the transition amplitude between a geodesic of size $b$ and a slice with $(L,\Phi)$ which is a finite cut-off trumpet. We perform the calculation semi-classically. For $L > b$, we find

$$\psi_b(L, \Phi = B\cosh(bx))$$
$$\approx \exp\left(-\frac{2\sqrt{L^2-b^2}}{b}|B|\sinh\left(\frac{b}{2}\right) + m\,\mathrm{arcsinh}\left(\frac{\sqrt{L^2-b^2}}{b}\right)\mathrm{sgn}(B)\right)\,. \tag{2.19}$$

As the classical solution of the closed universe has $L < b$, the regime of $L > b$ is classically forbidden. This is consistent with the fact that we see exponential decay in (2.19). For the classically allowed region $L < b$ we have

$$\psi_b(L, \Phi = B\cosh(bx)) \approx 2\cos\left(\frac{2\sqrt{b^2-L^2}}{b}B\sinh\left(\frac{b}{2}\right) - m\,\arcsin\left(\frac{\sqrt{b^2-L^2}}{b}\right)\right)\,, \tag{2.20}$$

and the oscillating behavior is as expected. One can check that the wave functions (2.19) and (2.20)

---

[9] This is similar to the wavefunction of the two sided BH computed in JT [26]. This results in the corner contribution to the extrinsic curvature being off-shell.

satisfy the Wheeler-de-Witt equation when $2B\frac{b}{L}\sinh\frac{b}{2} = m$.[10]

### Non-perturbative inner product in JT gravity.

For completeness and to give an example of the formalism outlined in the introduction, we briefly mention the non-perturbative inner product for the case of pure JT gravity. The Euclidean JT gravity action is given by[11]

$$I[g,\Phi] = -S_0\chi(M) - \frac{1}{2}\int \sqrt{g}\Phi(R+2)\,, \tag{2.21}$$

where $\chi$ is the Euler characteristic of the bulk manifold $M$ and suppresses higher genus contributions. We define the inner product $\langle b'|b\rangle$ to be computed by the Euclidean path integral with boundary conditions specified by geodesics of size $b, b'$

$$\langle b'|b\rangle = \int \mathcal{D}g\mathcal{D}\Phi e^{-I[g,\Phi]} = \quad + \dots. \tag{2.22}$$

States with $b \neq b'$ are orthogonal at the cylinder topology level. $\langle b'|b\rangle = \frac{1}{b}\delta(b-b') + \mathcal{O}(e^{-2S_0})$. However, as explained in the introduction the variance of such an inner product is large

$$|\langle b'|b\rangle|^2 = \langle b|b\rangle \times \langle b'|b'\rangle + \mathcal{O}(e^{-2S_0})\,. \tag{2.23}$$

This implies that the states $|b\rangle, |b'\rangle$ are not orthogonal. Running through the general argument in the introduction, we find the Hilbert space is one dimensional. All of these arguments extend to JT coupled to matter, where now states can be prepared by inserting operators of many flavors $i$ on the slices.

## 3 Non-perturbative aspects and a topological model

In section 2 we found that we could prepare distinct semi-classical states of closed universes by setting different asymptotic boundary conditions. In this section we will study non-perturbative corrections in a simple topological model that mimics JT coupled to a large number of matter fields of different flavors.

It would of course be more desirable to study both perturbative and non-perturbative aspects in the

---

[10] In principle the result of gravitational path integral should always satisfy WDW equation for any input $L$ and $\Phi(\sigma)$ [22]. Here we obtained (2.19) and (2.20) by saddle point approximation, which is only valid when $B$ is on shell.

[11] We do not need additional boundary terms for geodesic boundary conditions [36].

same model. However, the ensemble interpretation of JT gravity plus matter is still under investigation [37], and detailed calculations are much more involved without modifying the key conclusions.

## 3.1 A simple topological model

The topological model we consider is motivated by the Marolf-Maxfield model [20]. We define our topological model to consist of operators that create asymptotic circles with a flavor index $i$ that takes values in $i = 1, ..., k$.

$$|\psi_i\rangle \equiv \quad \underset{i}{\bigcirc} \qquad (3.1)$$

This definition is motivated by $\mathrm{tr}\big(e^{-\beta H} \mathcal{O}_i\big)$ in JT coupled to matter. We will think of these operators as creating states of closed universes.

We can consider performing the gravity path integral with some number of $|\psi_i\rangle$ boundaries inserted.[12] With our choice of action, an odd number of these insertions vanishes, and so we can always write our asymptotic boundary conditions as products of objects such as $\langle\psi_i|\psi_j\rangle$.

We define the Euclidean action on a manifold $M$ to be

$$I(M) = -S_0\chi + I_{\mathrm{matter}} \qquad (3.2)$$

where $\chi$ is the Euler characteristic, and $I_{\mathrm{matter}}$ is defined to pair up the flavor indices of the asymptotic circles $|\psi_i\rangle$ on which $M$ ends. The matter action multiplies the path integral by an integer given by the number of distinct pairings of flavors on $M$. This is motivated by $\mathrm{tr}\big(e^{-\beta H} \mathcal{O}_i\big) = 0$, and that for JT on a manifold all operator insertions $\mathcal{O}_i$ must connect to give a non-zero answer.

The simplest quantity we can consider in this model is the inner product matrix

$$M_{ij} \equiv \langle\psi_i|\psi_j\rangle \equiv \quad \begin{matrix} i \, \bigcirc \\[4pt] j \, \bigcirc \end{matrix} \, . \qquad (3.3)$$

We will study all moments of this matrix $M$. Let's examine the first few examples

$$\langle\psi_i|\psi_j\rangle = \delta_{ij} + \mathcal{O}(e^{-2S_0}), \qquad \qquad , \qquad (3.4)$$

---

[12] We do not consider insertions with multiple dots on a single circle.

$$|\langle\psi_i|\psi_j\rangle|^2 = 1 + \mathcal{O}(e^{-2S_0}), \qquad\qquad , \qquad i \neq j. \tag{3.5}$$

If we compare (3.4) and (3.5) we immediately notice that, as claimed in the introduction, we have $|\langle\psi_i|\psi_j\rangle|^2 \sim \langle\psi_i|\psi_i\rangle\langle\psi_j|\psi_j\rangle$. In other words, off-diagonal matrix elements are of the same order of magnitude as the diagonal elements.[13]

More generally, we have that $\mathrm{tr}(M^n) = (\mathrm{tr}\, M)^n$. As an example, consider $(\mathrm{tr}\, M)^3$. The boundary condition are given by [14]

$$(\mathrm{tr}\, M)^3 = \qquad\qquad . \tag{3.6}$$

In (3.6), the second row is identical to the first row. Next, we consider $\mathrm{tr}\, M^3$. It is given by

$$\mathrm{tr}\, M^3 = \qquad\qquad . \tag{3.7}$$

In (3.7), the second row is a permutation of the first row. Putting everything together, the boundary conditions in (3.6) is identical to that in (3.7). A Similar argument immediately shows the desired result $\mathrm{tr}(M^n) = (\mathrm{tr}\, M)^n$.

**Comparison to the black hole case.** It is useful to contrast the above with the calculation for a black hole where the rank of the Hilbert space is upper bounded by $e^{S_0}$ [21, 23, 40, 41]. The black hole carries an end of the world brane (with flavor $i$) which is attached to an asymptotically AdS Euclidean boundary. The inner products are [21]

$$\langle\psi_i|\psi_j\rangle \sim \delta_{ij}, \qquad\qquad , \qquad |\langle\psi_i|\psi_j\rangle|^2 \sim e^{-S_0}, \qquad\qquad . \tag{3.8}$$

Off-diagonal matrix elements are suppressed by $e^{-S_0/2}$ relative to the diagonal, and so we expect non-perturbative effects to not be important until late times or when one considers very complex

---

[13] Since these inner products are computed using the gravitational path integral they should be interpreted as ensemble averaged quantities [38, 39].

[14] In (3.6) and (3.7), contributions from repeated indices will be summed over.

quantities.

Let us see why $\mathrm{tr}(M^n) \neq \mathrm{tr}(M)^n$ for the black hole case, which allows for a larger dimensional Hilbert space. We again compare $(\mathrm{tr}\, M)^3$ and $\mathrm{tr}(M^3)$, we have

$$(3.9)$$

The presence of the asymptotic boundary crucially breaks the permutation symmetry of the boundary conditions, so the two quantities are clearly not equal. This is because the Cauchy slice defining the black hole state contain a boundary.

**Brief review of alpha states/sectors.** It was pointed out in [42–44] that including spacetime wormholes in the path integral is equivalent to an uncertainty in the coupling constants of the theory. The couplings are conventionally denoted by $\alpha$, and an $\alpha$-sector is when the couplings are fixed. In some special settings it has been shown that gravitational path integrals compute quantities that are averaged over different $\alpha$-sectors in an ensemble of theories. Well-known examples include pure JT gravity and its various variants [38]. In such theories, the gravity path integral evaluated with boundary conditions $Z$ is to be understood as computing

$$\langle Z \rangle = \int d\alpha P(\alpha) Z_\alpha \,. \tag{3.10}$$

In the above $Z_\alpha$ is the value of a partition function in the sector $\alpha$, and $P(\alpha)$ is a probability measure for distinct sectors.

**Connection to Marolf-Maxfield Model.** We briefly point out the connection of our topological model to the Marolf-Maxfield (MM) model [20].[15] In the MM model, the basic object is the operator $Z$ which creates an asymptotic boundary. We can evaluate $Z^n$ with $n \in \mathbb{N}$ in some state of closed universes. The simplest choice is the Hartle-Hawking state

$$\langle \mathrm{HH} | Z^n | \mathrm{HH} \rangle \,. \tag{3.11}$$

---

[15] We thank Douglas Stanford for emphasizing this point to us.

In our case, we can interpret the matrix $M_{ij}$ as inserting operators $Z_i, Z_j$ of flavored circles. The evaluation of our amplitude can then be written as

$$\langle \mathrm{HH} | Z_i Z_j | \mathrm{HH} \rangle \,, \tag{3.12}$$

with higher moments following from the above.

## 3.2   Ensemble averaging and the distribution of $\mathrm{tr}\, M$

In the previous subsection we came to two conclusions: non-perturbatively we have $\mathrm{rank}(M) = 1$ and that $\mathrm{tr}\, M$ should not be thought of as a fixed number but as a random variable evaluated over a probability distribution over $\alpha$-sectors

$$\langle \mathrm{tr}\, M \rangle = \int d\alpha P(\alpha) \, \mathrm{tr}\, M_\alpha \,. \tag{3.13}$$

Within each $\alpha$-sector $\mathrm{tr}\, M$ is given by a particular eigenvalue, but this eigenvalue can take different values in different members of the ensemble.

In this section we will evaluate this probability distribution in the special limit where we exclude higher genus effects by setting $e^{S_0} = \infty$. One way to extract the probability distribution for $\mathrm{tr}\, M$ is to compute all higher moments $\mathrm{tr}(M^n)$.[16] Since we are suppressing higher genus effects the only geometries that contribute are cylinders. The computation of $\mathrm{tr}\, M^n$ reduces to the problem of wick contractions of the same flavored cylinders. In this case we can easily solve for the probability distribution. We can represent each state by a zero-dimensional dot carrying flavor index $i$.

$$\tag{3.14}$$

$$\tag{3.15}$$

We assign each dot $i$ a random Gaussian variable $A_i$ with mean 0 and variance 1. $A_i$'s are independent for different $i$. Then we have the distribution

$$M_{ij} = \langle i | j \rangle = A_i A_j \,, \qquad \langle \mathrm{tr}\, M^n \rangle = \prod_{i=1}^{k} \int \frac{dA_i}{\sqrt{2\pi}} \exp\left(-\frac{A_i^2}{2}\right) \left(\sum_{i=1}^{k} A_i A_i\right)^n \,. \tag{3.16}$$

In gravity language, the above probability distribution wick contracts circles of the same flavor. We

---

[16] With the rules defined by the gravity path integral, bras and kets can connect with each other. This implies that the closed universe states live in a real Hilbert space [45]. More explicitly, to mimic the rules of the gravity path integral we find we can write states in an alpha sector as $A_{i,(\alpha)} | \alpha \rangle$, where the coefficients $A$ are real numbers. See equation (3.16).

see that fixing to an $\alpha$-sector amounts to fixing to a choice of values $A_{i,(\alpha)}$, and it's immediately clear that the matrix $M$ has rank one in each sector. Note that the above integral is identical to the model describing a completely evaporated black hole with $e^{S_{\text{BH}}} = 1$ [21]. In this sense there is some similarity between the closed universe and the black hole interior.

We find the distribution for $x = \text{tr}\, M$ is the chi-squared distribution

$$p_{\text{tr}\, M}(x) = \frac{x^{k/2-1}e^{-\frac{x}{2}}}{2^{\frac{k}{2}}\Gamma(\frac{k}{2})} \,. \tag{3.17}$$

The details are in appendix B.1.[17] From (3.17), the value of $\text{tr}\, M$ is centered at $k$ as expected. To compare the distributions at different value of $k$ we consider the normalized quantity $\frac{\text{tr}\, M}{k}$, finding

$$p^{(\infty)}(x) \equiv p_{\frac{\text{tr}\, M}{k}}(x) = \frac{x^{k/2-1}e^{-\frac{k}{2}x}}{\left(\frac{2}{k}\right)^{k/2}\Gamma(\frac{k}{2})} , \tag{3.18}$$

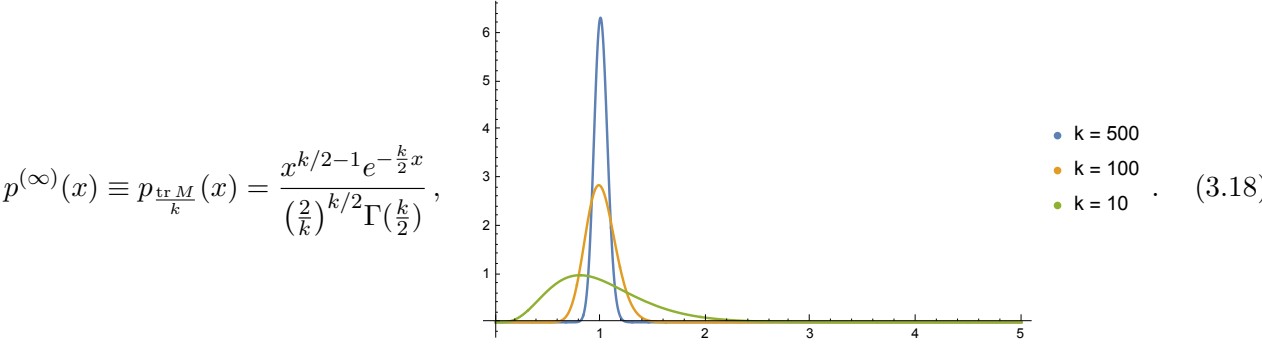

We denote this distribution by $p^{(\infty)}(x)$ as we will use it later. We see that as $k$ increases the distribution is more sharply peaked at 1, as we are washing out the variance in the square of the one-point function $A_i^2$.

**Semiclassical states from alpha sectors.** We would like to point out that the existence of the large number of semi-classical states we found is a result of a large number of $\alpha$-sectors.[18] In an alpha sector, the states can be labelled by $|\psi_i\rangle = A_{i,(\alpha)}|\alpha\rangle$, where $|\alpha\rangle$ is the unique state for the closed universe. The inner product matrix in each $\alpha$-sector is given by $N_{ij}^{(\alpha)} = A_{i,(\alpha)}A_{j,(\alpha)}$. Summing over $\alpha$ sectors, we obtain $N_{ij} = \sum_{l=1}^{m} A_{i,(\alpha_l)}A_{j,(\alpha_l)}$. We see the rank of this matrix is given by $\min(k, m)$, where $m$ is the number of alpha sectors and $k$ is the number of semiclassical states we tried to construct. In our case we could construct as many orthogonal semiclassical states as we wanted since our models have infinitely many alpha sectors.

---

[17] The chi-squared distribution can also be obtained directly by doing the Gaussian integral in (3.16).

[18] We thank Juan Maldacena for pointing this out.

### 3.2.1 Including higher topologies

In obtaining (3.17) we excluded higher topologies. We now include them by setting $e^{S_0}$ to be finite. The distribution we get for $\text{tr}\, M$ is given by

$$p_{\text{tr}\, M}(x) = e^{-\lambda}\delta(x) + \sum_{n=1}^{\infty} \frac{e^{-\lambda}\lambda^n}{n!}\left[\frac{e^{2S_0}}{kn}\frac{\left(\frac{k}{2}\right)^{\frac{k}{2}}}{\Gamma(\frac{k}{2})}\left(\frac{xe^{2S_0}}{kn}\right)^{\frac{k}{2}-1}e^{-\frac{k}{2}\frac{xe^{2S_0}}{kn}}\right]$$

$$= e^{-\lambda}\delta(x) + \sum_{n=1}^{\infty}\frac{e^{-\lambda}\lambda^n}{n!}p^{(\infty)}\left(\frac{xe^{2S_0}}{kn}\right)\frac{e^{2S_0}}{kn} \tag{3.19}$$

where $\lambda = \frac{e^{2S_0}}{1-e^{-2S_0}}$, and $p^{(\infty)}(x)$ was defined in equation (3.18). The derivation of (3.19) is in appendix B.2. From (3.19), we see that the distribution of $\text{tr}\, M$ is the convolution between the Poisson distribution with rate $\lambda$ and chi-squared distribution $p^{(\infty)}$. Since Poisson is discrete (3.19) is a sum of equally-spaced peaks (Figure 8).

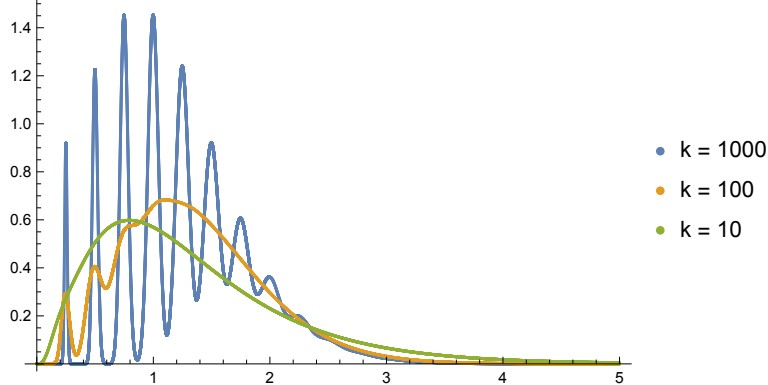

Figure 8: Probability distribution for $\frac{\text{tr}\, M}{k}$, with the inclusion of higher topologies, for different values of $k$ with $e^{2S_0} = 4$. We see that as $k$ gets larger, the peaks get sharper.

## 3.3 The meaning of $\text{tr}\, M$

We now discuss the interpretation of our probability distribution for $\text{tr}\, M$. One way to think about the model is that there are two independent $\alpha$ parameters we can tune. One parameter sets the number of states in the asymptotic boundary theory, which must be discrete. The second parameter, denoted $A_i$, gives values for the matter one point function evaluated on thermofield double state within each $\alpha$-sector.[19] In appendix B.2 we find that $\text{tr}\, M = Z\left(\sum_{i=1}^{k} A_i^2\right)$ as random variables. $Z$ satisfies Poisson

---

[19] As the distribution we obtain is continuous, it is not clear if the model satisfies reflection positivity as outlined in [46]. All of the states we considered have positive norm. We thank Xi Dong for discussion on this point.

statistics with rate $\lambda = \frac{e^{2S_0}}{1-e^{-2S_0}}$ and spacing $e^{-2S_0}$.[20] As a random variable, $Z$ can be defined by the following boundary condition:

$$Z \equiv \quad \bigg| \quad + \text{ higher genus } = 1 \cdot \frac{1}{1-e^{-2S_0}} \qquad (3.20)$$

$$Z^2 = \quad \bigg| \quad + \quad +\dots \qquad (3.21)$$

$$= 1 \cdot \left( \frac{1}{1-e^{-2S_0}} \right)^2 + e^{-2S_0} \cdot \frac{1}{1-e^{-2S_0}}.$$

The operator $Z$ is defined to be a pair of linked circles. When two circles are linked together, we are not allowed to separate them when forming higher topology surfaces. From this we can see that $Z$ is the trace of the identity operator over the Hilbert space of the two-sided wormhole, and higher genus effects show that it has a discrete spectrum.[21] We can thus identify

$$Z = \text{Tr}_{\mathcal{H}_{\text{WH}}}(\mathbb{1}). \qquad (3.22)$$

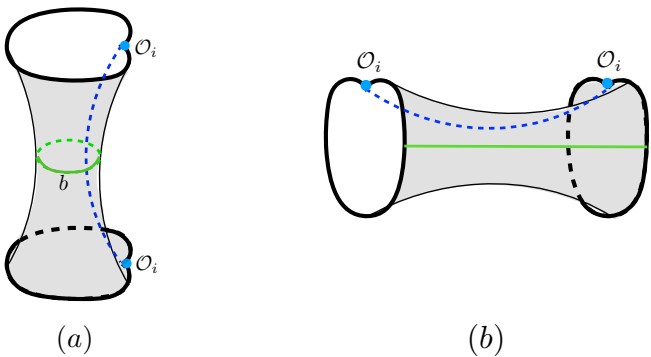

(a)                (b)

Figure 9: By cutting the double trumpet in different ways, we can obtain either a closed universe (a), or a two-sided wormhole (b).

    The appearance of two-sided wormholes is not a total surprise, as we can cut the double trumpet in

---

[20] More concretely, $Z \in e^{-2S_0} n$ with $n \in \mathbb{N}$ and the integer $n$ should be thought of as a boundary Hilbert space dimension. The prefactor can be adjusted by adding a boundary term to the topological action, see [20].

[21] The operator $Z$ we defined above is the analog of the operator $Z$ in Marolf-Maxfield model [20].

two ways. In one way we obtain the closed universe we have been discussing (Figure 9(a)). In the other way we get a two-sided wormhole (Figure 9(b)). We expect there will be some connection between the theory of closed universes and the theory of two-sided wormholes. It will be very interesting to understand this in more general contexts like JT coupled to matter, and we leave it to future work.

Putting everything together, we have

$$\operatorname{tr} M = \operatorname{Tr}_{\mathcal{H}_{\mathrm{WH}}}(\mathbb{1})\left(\sum_{i=1}^{k} A_i^2\right) \tag{3.23}$$

The theory of two-sided wormholes has a boundary dual, and so has a well-defined Hilbert space dimension within each $\alpha$-sector. The distribution we found with equally spaced peaks (3.19) arises from the discreteness of this Hilbert space dimension. The smearing around each peak comes from the average over $A_i$. We thus see

$$\operatorname{tr} M_\alpha = Z_\alpha\left(\sum_{i=1}^{k} A_{i,(\alpha)}^2\right), \tag{3.24}$$

where again in this model the first term on the right hand side of (3.24) gives the number of states in the corresponding two-sided boundary Hilbert space, whereas the second term gives the square of the thermal one-point function in an $\alpha$-sector.

More generally, the matrix elements of $M$ are given by

$$M_{ij} = Z(A_i A_j). \tag{3.25}$$

Where again (3.25) is an equality between random variables. The derivation closely follows from the derivation of (3.23) in appendix B.2.

# 4    Discussion

**One dimensional Hilbert space.**    A one dimensional closed universe Hilbert space and the existence of a single cosmological state is an old claim that has received renewed interest in recent years due to recent developments involving Euclidean wormholes [10,20,21,42,47–50]. The majority of the arguments rely on some form of the gravitational path integral, and so are restricted to a given gravitational theory. In [49] it was conjectured, through the baby universe hypothesis, that in string theory there is a unique closed universe state that unifies all closed universes across different spacetime dimensions.[22] This is a stronger claim regarding closed universes than the ones that can be argued for by the standard rules of the gravity path integral.

In this work we have compared perturbative and non-perturbative aspects of the theory of closed

---

[22] Abstractly, this unique state may be thought of as a wave-functional that takes as input boundary conditions corresponding to data on Cauchy slices across different spacetime dimensions with different fields on the slice.

universes in simple two dimensional models where we have complete control over the gravity path integral. We have found that the quantum gravity Hilbert space is one dimensional for closed universes. This arises from a choice of inner product defined by the gravity path integral,[23] first used in the context of dS$_2$ closed universes by [21] but also see [20,51]. We have further argued that this inner product will lead to a one dimensional closed universe Hilbert space beyond the simple models considered here.[24] These conclusions are heavily contrasted with the rich perturbative physics we find before including wormhole effects.

We end with some comments and questions, with the most crucial question being:

- **How do we describe the experience of an observer in a closed universe? How does Quantum Mechanics work with a single unique state?**

  We don't know. A semiclassical state with a massive worldline seems to be non-perturbatively equivalent to every other semiclassical state. All naively defined observables seem to receive large corrections from wormholes.

## 4.1   Comments on the unique cosmological state and tensor network

We showed that the gravitational path integral indicates that within each $\alpha$-sector there is a unique cosmological state of closed universes. A natural question is how to understand such a state. Let's fix one $\alpha$-sector and assume the boundary theory has a Hilbert space spanned by energy eigenstates $|E_i\rangle$. This closed cosmological state is *NOT* any state in the Hilbert space, that is it is not some linear combination of $|E_i\rangle$. In fact, it is a number. For example, $\text{tr}\left(e^{-\beta H}\mathcal{O}_i\right)$ is a number for fixed $H$ and $\mathcal{O}_i$. As all numbers are proportional to each other, there is a single closed universe state in each $\alpha$-sector.

How is it possible for a number to encode the rich semi-classical physics we discussed in section 2? It obviously cannot. In the discussion of section 2, we get the rich perturbative physics out of the boundary conditions we set at infinity. Different boundary conditions prepare different cosmological states, but at the end after taking trace all they give is a number in a fixed $\alpha$-sector. This number clearly contains very limited information, but how we obtained this number may encode rich physics.

For the simplest cases in the standard AdS/CFT dictionary, there is a one-to one correspondence between bulk and boundary states. Subtleties may already arise for complex enough states like late time black holes [52]. Our case of closed universes is more extreme, where the boundary state $\text{tr}\left(e^{-\beta H}\mathcal{O}_i\right)$ is simply a number in each $\alpha$-sector while the bulk contains rich physics.

---

[23] An obvious question is why this inner product is preferred over other choices. One answer is that when this inner product is applied to problems involving black holes in asymptotically AdS space we recover the correct black hole Hilbert space dimension $e^{S_0}$ [21,51].

[24] For more general theories in higher dimensional spacetimes additional assumptions need to be made regarding the gravitational path integral. The argument for a one-dimensional Hilbert space is more formal since moments of inner products need to be formally manipulated, and cannot be directly evaluated as can be done in two dimensions. See around equation (1.5) and footnote 2 for further discussion of the required assumptions.

In the language of tensor network [53, 54], in our case the output of the tensor network is simply a number. We certainly cannot reconstruct the tensor network out of one number, but it's the tensor network itself which encodes the semi-classical physics.

## 4.2 Connection to evaporating black hole and non-isometric codes

The closed universes considered in this paper are similar to the black hole interior. After the black hole completely evaporates, the radiation is in a pure state in a spacetime without a black hole, and the interior exists as a disconnected closed universe. This closed universe is inside the entanglement wedge of the radiation [21, 23, 39, 55].

In the language of non-isometric codes [52], there are two descriptions of the same state. In the semi-classical effective description, there is large amount of entanglement between the closed universe and the radiation. On the other hand, in the fundamental description, the radiation is in a pure state by itself. In order for this to happen, in the map from the effective description to the fundamental description, the seemingly different closed universe states will have to be mapped to numbers.

However, as discussed in [52], the map may have various problems when the evaporating black hole becomes too small as errors suppressed by $e^{-S_{\mathrm{BH}}}$ are no longer small. This is essentially the same problem as the contrast between rich semi-classical physics and unique state of closed universes. It seems likely that most naively defined observables in closed universes will receive large corrections from spacetime wormholes. How to understand this is an important problem.

## 4.3 The role of ensemble averaging in cosmology

The presence of $\alpha$-sectors has some appealing features when it comes to describing cosmology [56, 57]. One positive feature is the following. An observer in a universe might not exist long enough to fix all the couplings in the theory to reduce themselves to a single sector, and so it seems natural to average over theories consistent with their experience. We certainly do not know all aspects of the fundamental theory of our own universe, such as all coupling constants. Furthermore, in cosmologies that only exist for a finite amount of time there is a limit to the number of experiments that can be carried out to determine the $\alpha$-sector.

A possibility is that a unique $\alpha$-sector with a unique wavefunction for closed universes exists for quantum gravity [49]. This raises the problem of what observables we can measure when there is only one state.

It is important to mention that our JT closed universe construction depends on an ensemble averaging interpretation. The models we studied, JT coupled to matter, and the simple topological model, are both known to have ensemble duals. In our construction we had that $\mathrm{tr}\left(e^{-\beta H}\mathcal{O}_i\right) = 0$ but $\left[\mathrm{tr}\left(e^{-\beta H}\mathcal{O}_i\right)\right]^2 \neq 0$, which can only arise from an ensemble average. It is natural to ask how this story changes when we fix ourselves to a single theory. In a single $\alpha$-sector it is appealing to think that

$\mathrm{tr}\big(e^{-\beta H}\mathcal{O}_i\big)$ is no longer zero, and comes from some kind of bulk half-wormhole geometry, see [58, 59]. Such a contribution may then directly give us the state of the closed universe.

## 4.4   Comments on operator reconstruction

In the standard story of operator reconstruction in AdS/CFT, a particle in the bulk can be created in the entanglement wedge by acting with an operator in the boundary theory. In the case of a fully evaporated black hole, the black hole interior is a closed universe, and it is part of the entanglement wedge of the radiation. By applying unitary operators to the radiation one can create particles in the black hole interior.[25] In this case, the bulk dual (effective description) is more than just a closed universe. It is a closed universe entangled with the radiation, which is a well defined quantum system on which operators can act on.

In section 2 we defined states of the closed universes by specifying boundary conditions $\mathrm{tr}\big(e^{-\beta H}\mathcal{O}_i\big)$. This boundary condition prepares a state of a closed universe with a single particle. Clearly we can insert additional operators to change the state of the closed universe. For example, we can insert an additional particle by considering $\mathrm{tr}\big(e^{-\beta_1 H}\mathcal{O}_i e^{-(\beta-\beta_1)H}\mathcal{O}_j\big)$. At first sight this may look like we are applying a boundary operator to change the state. However, this operator reconstruction is quite different from the case of an evaporating black hole. We add excitations to the universe by changing boundary conditions, but the operator was applied before taking the trace. In the language of tensor networks, the operator was applied to the tensor network itself, not the state made by the tensor network.

## Acknowledgement

We thank Raphael Bousso, Yiming Chen, Xi Dong, Daniel Harlow, Juan Maldacena, Don Marolf, Geoff Penington, Xiaoliang Qi, Douglas Stanford, and Mark Van Raamsdonk for helpful discussions and comments. M.U. was supported in part by grant NSF PHY-2309135 to the Kavli Institute for Theoretical Physics (KITP), and by a grant from the Simons Foundation (Grant Number 994312, DG). Y.Z. was supported in part by the National Science Foundation under Grant No. NSF PHY-1748958 and by a grant from the Simons Foundation (815727, LB).

## A   Details on closed universes in JT+matter

We include some additional technical details not present in the main text.

---

[25] See [52] for potential subtleties.

## A.1 Pure JT gravity

**Classical solutions and symplectic form.** Given a Lagrangian density, the variation of the density takes the form

$$\delta L = E_a \delta \phi^a + d\Theta \,, \tag{A.1}$$

where $E_a$ are equations of motion and $d$ is a spacetime total derivative. For a closed universe, the second term vanishes when integrated over the spacetime manifold since there are no boundaries and so can be discarded. However it is necessary for computing the symplectic form on phase space [29]. The variation of the JT action can be taken and the pre-symplectic 1-form is given by

$$\Theta^\mu = \frac{1}{2}\sqrt{-g} \left( g^{\nu\alpha}g^{\mu\beta} - g^{\mu\nu}g^{\alpha\beta} \right) \left( \phi \nabla_\nu \delta g_{\alpha\beta} - \nabla_\nu \phi \delta g_{\alpha\beta} \right) \,. \tag{A.2}$$

The variation $\delta$ is an exterior derivative on phase space. In the main text we saw that the phase space was labelled by two parameters

$$ds^2 = -dt^2 + b^2 \cos^2(t) d\sigma^2, \qquad \Phi = \phi_c \sin(t) \,, \tag{A.3}$$

given by $b, \phi_c$ and so $\delta$ has the obvious action on the classical solutions. The current is defined by $\sqrt{-g}J^\mu = -\delta\Theta^\mu$ and it can be checked that it is conserved $\nabla_\mu J^\mu = 0$ so can be evaluated on any Cauchy slice. The symplectic 2-form is defined by

$$\omega = \int_\Sigma d\Sigma_\mu J^\mu \tag{A.4}$$

Evaluating the above on the classical solutions we find

$$\omega = \delta\phi_d \wedge \delta b \,, \tag{A.5}$$

as quoted in the main text.

**WdW wavefunction contours.** We elaborate on some contour calculations involving the WdW wavefunction. The wavefunctional was solved for dilaton gravity in [28, 30, 31], we primarily follow the notation of [31] where a careful Hamiltonian analysis of JT gravity is carried out. The case of most interest is for constant dilaton, repeated here

$$\Psi_{\phi_c}[\phi_0, L] = \begin{cases} \exp\left( iL\sqrt{\phi_c^2 - \phi_0^2 + i\epsilon} \right), & \phi_c > 0 \,, \\ \exp\left( -iL\sqrt{\phi_c^2 - \phi_0^2 - i\epsilon} \right), & \phi_c < 0 \,. \end{cases} \tag{A.6}$$

There are two branches of solutions to the WdW equation, and we find we must take different branches for $\phi_c > 0$ and $\phi_c < 0$ to get results consistent with semiclassical expectations. Additionally, to make

sense of the amplitude for $\phi_0 < \phi_c$ we must make a choice of an $i\epsilon$ prescription, which we have done above. With this definition the wavefunctionals agree at $\phi_c = 0$ as we approach from both sides. To transform the above into the geodesic length basis we use that $b, \phi_c$ are canonically conjugate so that $\langle \phi_c | b \rangle = \exp(-ib\phi_c)$. Transforming to the $b$ basis we will have that

$$\Psi_b[\phi_0, L] = \int d\phi_c \Psi[\phi_c; \phi_0, L] \langle \phi_c | b \rangle \tag{A.7}$$

The saddlepoint analysis is slightly subtle. For $b > L$ there are two saddles, for the positive/negative branchs we have $\phi_{c,\pm} = \pm b\phi_0/(b^2 - L^2)^{1/2}$. The saddles lie along the real axis with the saddle values away from the cuts in $\Psi$. For $L > b$, there are now two saddles along the imaginary axis at $\phi_{c,\pm} = \pm ib\phi_0/(L^2 - b^2)^{1/2}$. However, the contour for the negative branch cannot be deformed to run along the negative imaginary axis since the function exponentially grows at $-i|\phi_c|$. We can only pick up the saddle on the positive imaginary axis. This gives the oscillating and decaying behavior respectively in (2.7).

## A.2 Classical solution of closed universe with one observer

With vanishing one-point function: $\text{tr}\left(e^{-\beta H}\mathcal{O}_i\right) = 0$, we consider $|\text{tr}\left(e^{-\beta H}\mathcal{O}\right)_i|^2$. It's given by a wormhole as in figure 10. The insertion of the matter operator stabilizes the wormhole geodesic $b$. The insertion of the operator also backreacts and causes a corner in the schwarzian boundary. Here we will find the classical solution by imposing $SL(2)$ charge conservation at the insertion point of the particle, which amounts to putting the corner on-shell. The geometry can be constructed by cutting a piece out of the hyperbolic disk.

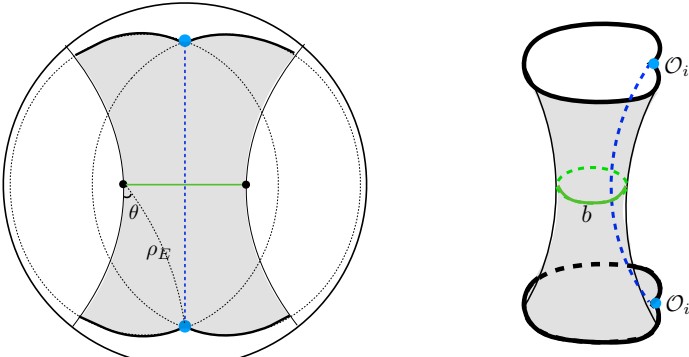

Figure 10: The wormhole geometry from a quotient of the hyperbolic disk. The particle is the blue line. The two thick black lines are geodesics and identified with each other. The green line is the geodesic $b$.

The metric on a hyperbolic disk is given by

$$ds^2 = d\rho^2 + \sinh^2 \rho \, d\phi^2 . \tag{A.8}$$

We construct the wormhole by creating two circles of size $\beta_E/\epsilon$ with radii $\rho_E$, and translating them in opposite directions on the disk. Their centers are given by the black dots in 10, and the black lines are geodesics passing through the circles. The actual asymptotic boundary has length $\beta/\epsilon$, and is a portion of the $\rho_E$ circles indicated by the thick black lines. Hyperbolic trigonometry applied to figure 10 gives us

$$2\pi \sinh \rho_E = \frac{\beta_E}{\epsilon}, \qquad \frac{\theta}{2\pi} = \frac{\beta}{2\beta_E}, \qquad \sin \theta = \frac{\tanh\left(\frac{b}{2}\right)}{\tanh \rho_E} \approx \tanh\left(\frac{b}{2}\right). \tag{A.9}$$

We need an additional equation equation which comes from $SL(2)$ charge conservation and puts the geometry on-shell. We work in embedding coordinates. The hyperbolic disk can be embedded in $\mathbb{R}^{1,2}$ with metric

$$-(X^0)^2 + (X^1)^2 + (X^2)^2 = -1, \qquad ds^2 = -(dX^0)^2 + (dX^1)^2 + (dX^2)^2. \tag{A.10}$$

First consider the charge associated to a circle. It is given by $\sqrt{2\phi_r E}(1,0,0) = 2\pi\frac{\phi_r}{\beta_E}(1,0,0)$. The charge associated with the geodesic of the observer is given by $m(0,0,1)$. Now we translate the circles in horizontal direction. It is given by a boost in $X^0 - X^2$ plane. Consider the plane $X^2 = vX^0$. The translated circle has center located at $(X^0, X^1, X^2) = (\frac{1}{\sqrt{1-v^2}}, 0, \frac{v}{\sqrt{1-v^2}})$. The distance to the origin $y$ satisfies $\cosh y = \frac{1}{\sqrt{1-v^2}} = \gamma$, so we have

$$T_2(x) = \begin{pmatrix} \cosh y & 0 & \sinh y \\ 0 & 1 & 0 \\ \sinh y & 0 & \cosh y \end{pmatrix} \tag{A.11}$$

Transforming the charge, we have

$$Q_r = \frac{2\pi\phi_r}{\beta_E}(\cosh y, 0, \sinh y), \quad Q_l = \frac{2\pi\phi_r}{\beta_E}(\cosh y, 0, -\sinh y), \tag{A.12}$$

Imposing charge conservation, we have

$$\frac{2\pi\phi_r}{\beta_E} 2\sinh y = m, \quad \sinh y = \frac{m\beta_E}{4\pi\phi_r}, \tag{A.13}$$

We arrive at the condition

$$\text{Charge conservation:} \quad \sinh\left(\frac{b}{2}\right) = \tan(\theta) = \tan\left(\frac{\pi\beta}{\beta_E}\right) = \frac{m\beta_E}{4\pi\phi_r}, \tag{A.14}$$

We consider the case where $m \gg \frac{\pi\phi_r}{\beta}$, $\beta_E \approx 2\beta$. We find the geometry satisfies

$$b_0 \approx 2\log\left(\frac{m\beta}{\pi\phi_r}\right). \tag{A.15}$$

We see that we can stabilize the geometry at a very large size $b_0$ provided we take the mass of the field

to be very large.

### A.2.1 dilaton profile

When matter is included the dilaton is also on-shell on the double trumpet geometry. We can extract the dilaton from the embedding into the disk, where the dilaton profile is manifest. We draw some embeddings in figure 11. The lines in Figure 11(c) are lines of constant dilaton.

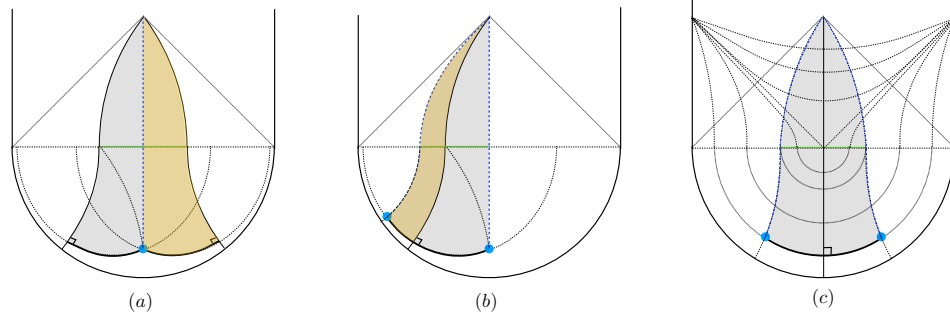

Figure 11: Embedding of the closed universe in the hyperbolic disk, with an analytic continuation at the time symmetric slice.

The coordinate and dilaton of the closed universe are given by

$$ds^2 = -dt^2 + \cos^2 t \, d\sigma^2 \,, \qquad \phi = \phi_h r = \phi_h \cosh \sigma \cos t \,. \tag{A.16}$$

Using that $r = \cosh \rho$, $2\pi r_c = \frac{\beta_E}{\epsilon}$, $\phi_h r_c = \frac{\phi_r}{\epsilon}$ we have that $\frac{\phi_h}{2\pi} = \frac{\phi_r}{\beta_E} \approx \frac{\phi_r}{2\beta}$. This immediately gives us

$$\phi(\sigma, t) = \frac{2\pi \phi_r}{\beta_E} \cos t \cosh \sigma \,. \tag{A.17}$$

### A.3 Classical solution with one observer and one light particle

Next, we consider the case where the two insertions are separated by general Euclidean time $\beta_1$ and $\beta_2$, $\beta_1 + \beta_2 = \beta$.

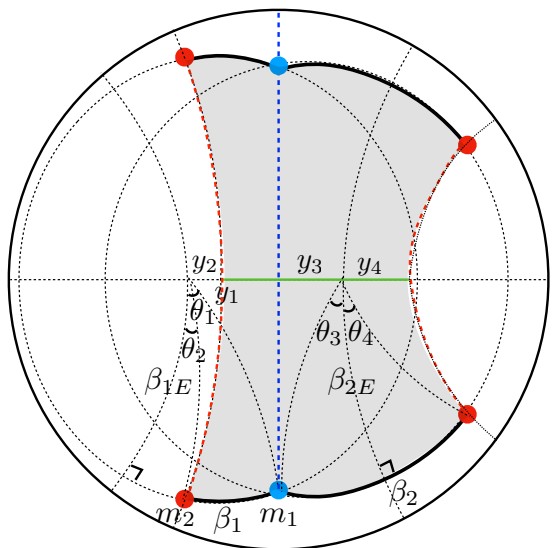

Figure 12: Embedding of the two-particle double trumpet into a hyperbolic disk.

From geometry, we have

$$\tan(\theta_i) = \sinh y_i, \quad \sin\theta_i = \tanh y_i, \quad \cos\theta_i = \frac{1}{\cosh y_i} \quad i = 1,2,3,4$$

$$\frac{\theta_1 + \theta_2}{2\pi} = \frac{\beta_1}{\beta_{1E}}$$

$$\frac{\theta_3 + \theta_4}{2\pi} = \frac{\beta_2}{\beta_{2E}}$$

From these we get

$$\tan\left(\frac{2\pi\beta_1}{\beta_{1E}}\right) = \frac{\sinh y_1 + \sinh y_2}{1 - \sinh y_1 \sinh y_2} \tag{A.18}$$

$$\tan\left(\frac{2\pi\beta_2}{\beta_{2E}}\right) = \frac{\sinh y_3 + \sinh y_4}{1 - \sinh y_3 \sinh y_4} \tag{A.19}$$

From charge conservation:

$$\frac{2\pi C}{\beta_{1E}}(\cosh y_1, \sinh y_1, 0) - \frac{2\pi C}{\beta_{2E}}(\cosh y_3, -\sinh y_3, 0) = m_1(0,1,0)$$

From this, we first get that

$$\frac{\beta_{1E}}{\beta_{2E}} = \frac{\cosh y_1}{\cosh y_3} = \frac{\cosh y_2}{\cosh y_4} \tag{A.20}$$

$$\frac{\sinh y_1}{\beta_{1E}} + \frac{\sinh y_3}{\beta_{2E}} = \frac{m_1}{2\pi\phi_r} \tag{A.21}$$

$$\frac{\sinh y_2}{\beta_{1E}} + \frac{\sinh y_4}{\beta_{2E}} = \frac{m_2}{2\pi\phi_r} \tag{A.22}$$

From equations (A.18),(A.19),(A.20),(A.21),(A.22), we have six unknowns and six equations. For consistency, we need to check that the red lines on the left and right have the same length. We need

$$\frac{\beta_{1E}}{\beta_{2E}} = \frac{\cos\theta_3}{\cos\theta_1} = \frac{\cos\theta_4}{\cos\theta_2}$$

Notice that this is automatically satisfied if the equations of motion are satisfied.

We work in the limit where $m_1 \gg \frac{\pi\phi_r}{\beta} \gg m_2$. We have $\theta_1 \approx \frac{\pi}{2}$, $\theta_3 \approx \frac{\pi}{2}$. Then $y_1$ and $y_3$ will be large.

$$\frac{\beta_{1E}}{\beta_{2E}} = \frac{\cos\theta_4}{\cos\theta_2} \approx \frac{\cos\left(\frac{2\pi\beta_2}{\beta_{2E}} - \frac{\pi}{2}\right)}{\cos\left(\frac{2\pi\beta_1}{\beta_{1E}} - \frac{\pi}{2}\right)} = \frac{\sin\left(\frac{2\pi\beta_2}{\beta_{2E}}\right)}{\sin\left(\frac{2\pi\beta_1}{\beta_{1E}}\right)}$$

$$\sinh y_2 \approx -\frac{1}{\tan\left(\frac{2\pi\beta_1}{\beta_{1E}}\right)}$$

$$\sinh y_4 \approx -\frac{1}{\tan\left(\frac{2\pi\beta_2}{\beta_{2E}}\right)}$$

The charge conservation becomes

$$\frac{e^{y_3}}{\beta_{2E}} \approx \frac{m_1}{2\pi C} \approx \frac{e^{y_1}}{\beta_{1E}}$$

$$\frac{1}{\beta_{1E}} \frac{1}{\tan\left(\frac{2\pi\beta_1}{\beta_{1E}}\right)} + \frac{1}{\beta_{2E}} \frac{1}{\tan\left(\frac{2\pi\beta_2}{\beta_{2E}}\right)} = -\frac{m_2}{2\pi\phi_r}$$

Let $\frac{2\pi\beta_1}{\beta_{1E}} = x_1$, $\frac{2\pi\beta_2}{\beta_{2E}} = x_2$, we have

$$\frac{\beta_1}{\beta_2} \frac{x_2}{x_1} = \frac{\sin x_2}{\sin x_1}$$

$$\frac{1}{\beta_1} \frac{x_1}{\tan x_1} + \frac{1}{\beta_2} \frac{x_2}{\tan x_2} = -\frac{m_2}{\phi_r}$$

From these one can solve for $\beta_{1E}$ and $\beta_{2E}$. We rewrite the above equations as

$$\cos x_1 + \cos x_2 = -\frac{m_2\beta}{\pi\phi_r} \frac{\pi\beta_2 \sin x_2}{\beta x_2} = -\frac{m_2\beta}{\pi\phi_r} \frac{\pi\beta_1 \sin x_1}{\beta x_1}$$

Let's first solve for the case when $m_2 = 0$. We have

$$x_1 + x_2 = \pi, \quad \frac{x_1}{x_2} = \frac{\beta_1}{\beta_2}$$

$$x_1 = \pi\frac{\beta_1}{\beta}, \quad x_2 = \pi\frac{\beta_2}{\beta}$$

$$\beta_{1E} = \beta_{2E} = 2\beta$$

Note that $y_2 + y_4 = 0$. One of them is positive and one of them is negative.

Next we solve it to linear order in $\frac{m_2\beta}{\pi C} \equiv \epsilon$. We let

$$x_1 = \pi\frac{\beta_1}{\beta} + z_1\epsilon, \quad x_2 = \pi\frac{\beta_2}{\beta} + z_2\epsilon$$

We have

$$\frac{\beta_2}{\beta}\frac{\sin\frac{\pi\beta_2}{\beta} + z_2\epsilon\cos\frac{\pi\beta_2}{\beta}}{\frac{\pi\beta_2}{\beta} + z_2\epsilon} = \frac{\beta_1}{\beta}\frac{\sin\frac{\pi\beta_1}{\beta} + z_1\epsilon\cos\frac{\pi\beta_1}{\beta}}{\frac{\pi\beta_2}{\beta} + z_1\epsilon}$$

$$1 + z_2\epsilon\left(\frac{1}{\tan\frac{\pi\beta_2}{\beta}} - \frac{1}{\frac{\pi\beta_2}{\beta}}\right) = 1 + z_1\epsilon\left(\frac{1}{\tan\frac{\pi\beta_1}{\beta}} - \frac{1}{\frac{\pi\beta_1}{\beta}}\right)$$

$$z_1 = \frac{1}{\pi}\frac{\pi\left(\frac{1}{\tan\frac{\pi\beta_2}{\beta}} - \frac{1}{\frac{\pi\beta_2}{\beta}}\right)}{\left(\frac{1}{\tan\frac{\pi\beta_2}{\beta}} - \frac{1}{\frac{\pi\beta_2}{\beta}}\right) + \left(\frac{1}{\tan\frac{\pi\beta_1}{\beta}} - \frac{1}{\frac{\pi\beta_1}{\beta}}\right)} = \frac{1}{\pi}\frac{\pi\beta_1}{\beta}\left(1 - \frac{\frac{\pi\beta_2}{\beta}}{\tan\left(\frac{\pi\beta_2}{\beta}\right)}\right)$$

$$z_2 = \frac{1}{\pi}\frac{\pi\left(\frac{1}{\tan\frac{\pi\beta_1}{\beta}} - \frac{1}{\frac{\pi\beta_1}{\beta}}\right)}{\left(\frac{1}{\tan\frac{\pi\beta_2}{\beta}} - \frac{1}{\frac{\pi\beta_2}{\beta}}\right) + \left(\frac{1}{\tan\frac{\pi\beta_1}{\beta}} - \frac{1}{\frac{\pi\beta_1}{\beta}}\right)} = \frac{1}{\pi}\frac{\pi\beta_2}{\beta}\left(1 - \frac{\frac{\pi\beta_1}{\beta}}{\tan\left(\frac{\pi\beta_1}{\beta}\right)}\right)$$

Now we assume $\beta_1 < \beta_2$, we have $x_1 < \frac{\pi}{2}$ and $x_2 > \frac{\pi}{2}$. As a result, $y_2 < 0$ and $y_4 > 0$. That's why we draw the figure as in Figure 12.

As a result,

$$x_1 = \frac{\pi\beta_1}{\beta}\left(1 + \frac{\epsilon}{\pi}\left(1 - \frac{\frac{\pi\beta_2}{\beta}}{\tan\left(\frac{\pi\beta_2}{\beta}\right)}\right)\right)$$

$$x_2 = \frac{\pi\beta_2}{\beta}\left(1 + \frac{\epsilon}{\pi}\left(1 - \frac{\frac{\pi\beta_1}{\beta}}{\tan\left(\frac{\pi\beta_1}{\beta}\right)}\right)\right)$$

We have

$$\beta_{1E} = 2\beta\left(1 - \frac{m_2\beta}{\pi^2 C}\left(1 - \frac{\frac{\pi\beta_2}{\beta}}{\tan\left(\frac{\pi\beta_2}{\beta}\right)}\right)\right)$$

$$\beta_{2E} = 2\beta \left( 1 - \frac{m_2 \beta}{\pi^2 C} \left( 1 - \frac{\frac{\pi \beta_1}{\beta}}{\tan\left(\frac{\pi \beta_1}{\beta}\right)} \right) \right)$$

$$y_1 = \log\left(\frac{m_1 \beta}{\pi \phi_r}\right) - \frac{m_2 \beta}{\pi^2 \phi_r}\left( 1 - \frac{\frac{\pi \beta_2}{\beta}}{\tan\left(\frac{\pi \beta_2}{\beta}\right)} \right)$$

$$y_3 = \log\left(\frac{m_1 \beta}{\pi \phi_r}\right) - \frac{m_2 \beta}{\pi^2 \phi_r}\left( 1 - \frac{\frac{\pi \beta_1}{\beta}}{\tan\left(\frac{\pi \beta_1}{\beta}\right)} \right)$$

$$y_2 = -\operatorname{arcsinh}\left( \frac{1}{\tan\left(\pi \frac{\beta_1}{\beta}\right) + \frac{1}{\cos^2 \frac{\pi \beta_1}{\beta}} z_1 \epsilon} \right)$$

$$= -\operatorname{arcsinh}\left( \frac{1}{\tan\left(\frac{\pi \beta_1}{\beta}\right)} \right) + \frac{1}{\sin\left(\frac{\pi \beta_1}{\beta}\right)} \frac{m_2 \beta}{\pi^2 \phi_r} \frac{\pi \beta_1}{\beta}\left( 1 - \frac{\frac{\pi \beta_2}{\beta}}{\tan\left(\frac{\pi \beta_2}{\beta}\right)} \right)$$

$$y_4 = -\operatorname{arcsinh}\left( \frac{1}{\tan\left(\frac{\pi \beta_2}{\beta}\right) + \frac{1}{\cos^2 \frac{\pi \beta_2}{\beta}} \epsilon z_2} \right)$$

$$= -\operatorname{arcsinh}\left( \frac{1}{\tan\left(\frac{\pi \beta_2}{\beta}\right)} \right) + \frac{1}{\sin \frac{\pi \beta_2}{\beta}} \frac{m_2 \beta}{\pi^2 \phi_r} \frac{\pi \beta_2}{\beta}\left( 1 - \frac{\frac{\pi \beta_1}{\beta}}{\tan\left(\frac{\pi \beta_1}{\beta}\right)} \right)$$

We have

$$b = y_1 + y_2 + y_3 + y_4$$

$$\approx 2\log\left(\frac{m_1 \beta}{\pi \phi_r}\right) + \frac{m_2 \beta}{\pi^2 \phi_r}\left[ \left( \frac{\frac{\pi \beta_1}{\beta}}{\sin\left(\frac{\pi \beta_1}{\beta}\right)} - 1 \right)\left( 1 - \frac{\frac{\pi \beta_2}{\beta}}{\tan\left(\frac{\pi \beta_2}{\beta}\right)} \right) + \left( \frac{\frac{\pi \beta_2}{\beta}}{\sin\left(\frac{\pi \beta_2}{\beta}\right)} - 1 \right)\left( 1 - \frac{\frac{\pi \beta_1}{\beta}}{\tan\left(\frac{\pi \beta_1}{\beta}\right)} \right) \right]$$

$$= b_0 + \frac{m_2 \beta}{\pi \phi_r}\left[ \frac{1 - \cos\left(\frac{\pi \beta_1}{\beta}\right)}{\sin\left(\frac{\pi \beta_1}{\beta}\right)} - \frac{2}{\pi}\left( 1 - \frac{\frac{\pi \beta_1}{\beta}}{\tan\left(\frac{\pi \beta_1}{\beta}\right)} \right) \right]$$

We noticed that $b$ will depend on $\beta_1$, and achieves maximal value when $\beta_1 = \beta_2 = \frac{\beta}{2}$.

Consider the dilaton profile. It is given by

$$\phi(t, x) = \begin{cases} \frac{2\pi \phi_r}{\beta_{2E}} \cos t \cosh(x - y_3) & 0 < x < y_3 + y_4 \\ \frac{2\pi \phi_r}{\beta_{1E}} \cos t \cosh(x + y_1) & -(y_1 + y_2) < x < 0 \end{cases} \tag{A.23}$$

It's easy to check that the dilaton is continuous across the perturbation. Its derivative is discontinuous as a result of the stress-energy tensor of the perturbation.

## A.4 Off-shell wormhole Computation

We now construct the off-shell wormhole solution in JT coupled to matter, see [33–35] for closely related constructions. The correlator we want to evaluate is $\left(\text{Tr}[e^{-\beta H}\mathcal{O}_i]\right)^2$ where $\mathcal{O}_i$ is a matter insertion of a massive field with mass $m$. In Euclidean signature the action with asymptotic boundaries is

$$I = -\frac{1}{2}\int \sqrt{g}\Phi(R+2) - \int_{\partial M} \sqrt{h}\Phi_b\left(K-1\right) + I_{\text{m}}\,, \tag{A.24}$$

where $I_m$ is the matter action of a massive free scalar. We take boundary conditions where the boundaries take fixed lengths $\beta/\epsilon$ and the dilaton is fixed to be $\Phi_b = \phi_r/\epsilon$. Since the one-point function of matter fields vanishes on the disk $\langle\mathcal{O}_i\rangle = 0$ the dominant geometry is the double trumpet. For a sufficiently massive particle the geodesic approximation holds and suppresses the wormhole as $\mathcal{O}_i\mathcal{O}_i \sim e^{-mL}$ where $L$ is the geodesic distance between the operator insertions.

On the double trumpet, the full expression for the two sided two point function is given by

$$\left(\text{Tr}[e^{-\beta H}\mathcal{O}_i]\right)^2 = \underbrace{\int_0^\infty db \int_0^b d\tau}_{\text{Moduli: length, twist}} \underbrace{\int \mathcal{D}f_L \mathcal{D}f_R e^{-I_{\text{sch}}[f_L]-I_{\text{sch}}[f_R]}}_{\text{Schw. modes}} \underbrace{Z_m(b)}_{\text{matter one-loop det}} \tag{A.25}$$

$$\times \underbrace{\sum_{w=-\infty}^{\infty}\left(\frac{f'_L(\tau_{j,L})f'_R(\tau_{j,R})}{\cosh^2\left[\frac{b}{2\beta}\left(f_L(\tau_{j,L}) - f_R(\tau_{j,R} + w\beta)\right)\right]}\right)^m}_{\substack{\text{two-pt function dressed to schw.}\\ \text{including winding } w}}. \tag{A.26}$$

The operators are inserted with a relative twist $\tau$ with respect to each other at times $\tau_R = \tau_L + \tau$. Here $f$ are left and right Schwarzian profiles. We restrict ourselves to the dominant configuration where the operators are inserted opposite each other, so that $\tau = 0$. We also ignore the matter one-loop determinant, which is given by the Selberg zeta function on the double trumpet, and comes from worldlines wrapping the closed $b$ cycle. This gives a subleading contribution when the wormhole $b$ is large.[26] On the second line we have the two point function dressed to the Schwarzian boundary [61], at large $m$ this backreacts on the classical profiles $f_L, f_R$. We restrict ourselves to the non-winding geodesic $w = 0$.

After these restrictions the classical action is given by

$$I = -\int \sqrt{h}\Phi_b(K-1) + m\log\left(2\frac{\epsilon^2}{\phi_r^2}\cosh D\right). \tag{A.27}$$

where the integral is over two boundaries, and the matter contribution is evaluated with the renormalized length $L = L_{\text{bare}} - 2\log\Phi_b$. Away from the corner where the matter operators are inserted we

---

[26] At small $b$ this gives a universal divergence $\lim_{b\to 0} Z_m(b) \sim e^{\frac{\pi^2}{6b}}$. Our analysis is thus not valid at small $b$ since we do not include winding geodesics connecting the two operator insertions. It was argued in [60] that the summation over winding geodesics stabilizies the wormhole against this divergence.

take the equations of motion to hold, and so the boundary cutout is of constant extrinsic curvature except at the location of the operator insertion. We allow for the geometry to be slight off shell by not enforcing the equations of motion at the corner.

The three contributions to the action consist of: the particle worldline, the on-shell schwarzian, the off-shell corner term. To evaluate each contribution we need some basic hyperbolic geometry identities, with the geometry denoted in figure A.4.

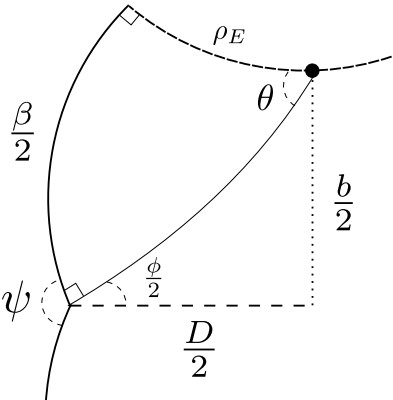

Figure 13: Geometric construction of the double trumpet with important angles and distances.

The basic identities of hyperbolic geometry give us

$$\cos\theta = \frac{\sinh\frac{D}{2}}{\sinh\rho_E}, \qquad \sin\theta = \frac{\tanh\frac{b}{2}}{\tanh\rho_E}, \qquad \cot\theta = \frac{\tanh\frac{D}{2}}{\sinh\frac{b}{2}}, \tag{A.28}$$

$$\tan\theta = \tan\left(\frac{\pi\beta}{\beta_E}\right) = \sinh\frac{b}{2}, \qquad \cos\theta = \frac{1}{\cosh\frac{b}{2}}, \tag{A.29}$$

$$\cos\frac{\phi}{2} = \frac{\tanh\frac{D}{2}}{\tanh\rho_E}, \qquad \sin\frac{\phi}{2} = \frac{\sinh\frac{b}{2}}{\sinh\rho_E}. \tag{A.30}$$

where we also have that $\sinh\rho_E = \frac{\beta_E}{2\pi\epsilon}$ and $\theta = \frac{\pi\beta}{\beta_E}$, along with $\psi = \pi - \phi$. Combining the first equation with the answer for $\theta$, and expanding for small $\epsilon$ we find the geodesic distance

$$D = 2\log\left(\frac{\beta_E}{\pi\epsilon}\cos\left(\frac{\pi\beta}{\beta_E}\right)\right), \qquad \cosh D = \frac{\beta_E^2}{2\pi^2\epsilon^2}\cos\left(\frac{\pi\beta}{\beta_E}\right). \tag{A.31}$$

The angle $\phi$ can similarly be found by using the equation

$$\sin\left(\frac{\phi}{2}\right) = \frac{2\pi\epsilon}{\beta_E}\sinh\left(\frac{b}{2}\right) \Rightarrow \phi = \frac{4\pi\epsilon}{\beta_E}\tan\left(\frac{\pi\beta}{\beta_E}\right). \tag{A.32}$$

**Boundary term.** The Schwarzian boundaries are at constant radius $\rho_E$ for a total length $\beta$. At constant radius $\rho_E$, we have extrinsic curvature $K = \coth\rho_E$. The boundary term, excluding the

corner, thus contributes

$$I_{\text{bdy}} \supset -\frac{2\phi_r\beta}{\epsilon^2}\left(\coth\rho_E - 1\right) = -\frac{4\pi^2\phi_r\beta}{\beta_E^2}\,. \tag{A.33}$$

where the second term is from $\rho_E = \log\frac{\beta_E}{\pi\epsilon}$. The corner contributes a delta function to the extrinsic curvature proportional to the deficit angle

$$\int_{\text{corner}} ds\sqrt{h}K = \pi - \psi\,, \tag{A.34}$$

where we defined $\psi$ above. We previously found that the angle $\pi - \psi = \frac{4\pi\epsilon}{\beta_E}\tan\left(\frac{\pi\beta}{\beta_E}\right)$. Note that this is order $\epsilon$, but it cancels the $\epsilon$ divergence in the dilaton, we thus get the total contribution from two corners

$$I_{\text{corner}} = \frac{8\pi\phi_r}{\beta_E}\tan\left(\frac{\pi\beta}{\beta_E}\right)\,. \tag{A.35}$$

The total boundary action is thus

$$I_{\text{bdy}} = -4\pi^2\phi_r\frac{\beta}{\beta_E^2} + \frac{8\pi\phi_r}{\beta_E}\tan\left(\frac{\pi\beta}{\beta_E}\right)\,. \tag{A.36}$$

**Particle Action.** The geodesic action is simple to compute since we already found $D$

$$I_{\text{particle}} = m\log\left(2\frac{\epsilon^2}{\phi_r^2}\cosh D\right) = 2m\log\left(\frac{\beta_E\cos\left(\frac{\pi\beta}{\beta_E}\right)}{\pi\phi_r}\right)\,. \tag{A.37}$$

**Total action.** We thus arrive at the total action

$$I = 2m\log\left(\frac{\beta_E\cos\left(\frac{\pi\beta}{\beta_E}\right)}{\pi\phi_r}\right) - 4\pi^2\phi_r\frac{\beta}{\beta_E^2} + \frac{8\pi\phi_r}{\beta_E}\tan\left(\frac{\pi\beta}{\beta_E}\right)\,. \tag{A.38}$$

The saddlepoint is given by

$$\partial_{\beta_E}I = 0 \implies \tan\frac{\pi\beta}{\beta_E} = \frac{m\beta_E}{4\pi\phi_r}. \tag{A.39}$$

This is the same condition we found from charge conservation in section A.2. We see at this value the Schwarzian is on-shell, including the corner. We can calculate the on-shell action

$$I_{\text{on-shell}} = 2m(1 - \log\frac{m}{4}) - \frac{\pi^2\phi_r}{\beta}\,, \tag{A.40}$$

where we have expanded in large $m$.

**Action in $b$ basis.** Using the identities at the start of the section, we can rewrite $\beta_E$ in terms of $b$ and express the action in terms of the wormhole size

$$I = 2m \log \left( \frac{\beta/\phi_r}{\cosh\left(\frac{b}{2}\right) \arctan\left(\sinh\frac{b}{2}\right)} \right) - \frac{4\phi_r}{\beta} \arctan^2\left(\sinh\frac{b}{2}\right) + \frac{8\phi_r}{\beta} \sinh\left(\frac{b}{2}\right) \arctan\left(\sinh\frac{b}{2}\right) \tag{A.41}$$

The saddlepoint satisfies

$$\sinh\left(\frac{b}{2}\right) \arctan\left(\sinh\left(\frac{b}{2}\right)\right) = \frac{m\beta}{4\phi_r} . \tag{A.42}$$

At large $m$ the solution is given by

$$b \approx 2 \log \left( \frac{\beta m + 4\phi_r}{\pi \phi_r} \right) . \tag{A.43}$$

This again matches the result we obtained in section A.2.

## A.5   Wave function on a generic slice

We obtained $\Psi_\beta(b)$, but we would like to transform to the basis where we can ask about more general Cauchy slices. The wavefunctional will be given by

$$\Psi_\beta[L, \phi] = \int db \Psi_\beta(b) \psi_b[L, \phi]$$

where $\Psi_\beta(b)$ was given by (2.17). In this section, we will obtain $\psi_b[L, \phi]$ by first going from the $b$ to the $L, K$ basis, and then transforming to the $L, \phi$ basis. We will do the computation by saddle point approximation. The Euclidean saddle is the double trumpet geometry with a matter excitation

$$ds^2 = d\rho^2 + b_0^2 \cosh^2(\rho) d\sigma^2 , \qquad \Phi(\rho, \sigma) \approx \frac{\pi \phi_r}{\beta} \cosh(\rho) \cosh(b\sigma) . \tag{A.44}$$

We first assume we are interested in a slice with $L > b$ with dilaton profile $\Phi = B \cosh(b\sigma)$. We must calculate the action between the geodesic throat and the slice with this data. On the trumpet geometry (A.44) there are two such slices with opposite extrinsic curvatures. The extrinsic curvature and length of constant $\rho$ slices, along with the action up to the slices, is given by

$$K = \tanh \rho , \quad L = b \cosh \rho , \quad b \sinh \rho = \sqrt{L^2 - b^2} \, \text{sgn}(\rho) , \tag{A.45}$$

$$I_{\text{grav}} = - \int d\sigma B \cosh(b\sigma) b \cosh(\rho) \tanh(\rho) = -\frac{2}{b} \sinh\left(\frac{b}{2}\right) B \sqrt{L^2 - b^2} \, \text{sgn}(\rho) , \tag{A.46}$$

$$I_{\text{m}} = m\rho = m \, \text{arcsinh}\left( \frac{\sqrt{L^2 - b^2}}{b} \right) \text{sgn}(\rho) , \tag{A.47}$$

where the sign of the matter action comes from the need to subtract the particle worldline action when

going backwards in time.

It is easier to first go to constant $L, K$ slices. The amplitude for the $b \to (L, K)$ cylinder is proportional to a delta function [36],[27] including the matter action

$$\psi_b(L, K) \sim \delta(K \mp \sqrt{1 - \frac{b^2}{L^2}}) \exp\left( \mp m \operatorname{arcsinh}\left( \frac{\sqrt{L^2 - b^2}}{b} \right) \right), \tag{A.48}$$

where the $\mp$ notation means we include a summation over both terms. Since $K$ and $\Phi$ are conjugate variables, we can perform the fourier transform to obtain the wavefunction at fixed $\Phi$

$$\psi_b(L, \Phi = B \cosh(bx)) \sim \int_C dK_E \exp\left( \int d\sigma \sqrt{\gamma} K_E \Phi \right) \psi_b(L, K_E)$$

$$\sim \int_C dK_E \exp\left( \frac{2L}{b} K_E B \sinh\left( \frac{b}{2} \right) \right) \delta(K_E \mp \sqrt{1 - \frac{b^2}{L^2}}) \exp\left( \mp m \operatorname{arcsinh}\left( \frac{\sqrt{L^2 - b^2}}{b} \right) \right). \tag{A.49}$$

where we emphasize that in these formulas $K_E$ is the euclidean extrinsic curvature. We must deform along the steepest descent contour. Notice that at real infinite $K_E$ we need $K_E B < 0$. So we can only pick up one of the delta function contributions

$$\psi_b(L > b, \Phi = B \cosh(bx)) \sim \exp\left( -\left[ 2\frac{\sqrt{L^2 - b^2}}{b} |B| \sinh\left( \frac{b}{2} \right) - m \operatorname{arcsinh}\left( \frac{\sqrt{L^2 - b^2}}{b} \right) \operatorname{sgn}(B) \right] \right)$$

Note that for large $L$, the first term always dominates, which is exponentially decreasing.

Next, we consider the case when $L < b$. In this case there does not exist a Cauchy slice in the Euclidean geometry with such data, and we must deform the double trumpet (A.44) into the complex $\rho = it$ plane to see the closed universe with such data. Analytically continuing, the quantities on the Lorentzian geometry become

$$L = b \cos t, \quad |t| = \arcsin \frac{\sqrt{b^2 - L^2}}{b}, \quad K_E = -i K_L \tag{A.50}$$

$$\exp(i I_{m,L}) = \exp\left( -i m \arcsin\left( \frac{\sqrt{b^2 - L^2}}{b} \right) \operatorname{sgn}(t) \right), \quad K_L = -\tan t. \tag{A.51}$$

The amplitude to go from the geodesic $b$ into the Lorentzian geometry is thus

$$\psi_b(L, K_L) = \delta(K_L \mp \frac{\sqrt{b^2 - L^2}}{L}) \exp\left( \pm i m \arcsin\left( \frac{\sqrt{b^2 - L^2}}{b} \right) \right). \tag{A.52}$$

(A.52) can also be seen from the Euclidean answer (A.48). The transform to the $\phi$ basis can be done

---

[27] There is also a one-loop determinant factor that we do not include since we only work with the semiclassical action.

either with the Lorentzian or the Euclidean signature. In either case, we find

$$\psi_b(L < b, \Phi = B\cosh(bx)) \sim \int_{-\infty}^{+\infty} dK_L \exp\left(-i\int dx\sqrt{\gamma}K_L\phi\right)\psi_b(L, K_L)$$

$$= \int_{-\infty}^{+\infty} dK_L \exp\left(-iLK_L\frac{2B}{b}\sinh\frac{b}{2}\right)\delta(K_L \mp \frac{\sqrt{b^2 - L^2}}{L})\exp\left(\pm im\arcsin\left(\frac{\sqrt{b^2 - L^2}}{b}\right)\right)$$

$$= \exp\left(-2i\frac{\sqrt{b^2 - L^2}}{b}B\sinh\frac{b}{2} + im\arcsin\left(\frac{\sqrt{b^2 - L^2}}{b}\right)\right) + c.c.$$

$$= 2\cos\left(2\frac{\sqrt{b^2 - L^2}}{b}B\sinh\frac{b}{2} - m\arcsin\left(\frac{\sqrt{b^2 - L^2}}{b}\right)\right). \tag{A.53}$$

When we set $L < b$, the delta function in (A.52) gives two complex conjugate contributions which are both picked up with the standard contour, and so we get real and oscillating behavior.

# B  Distribution of norm squared of the cosmological state

## B.1  Distribution of $\operatorname{tr} M$ when $e^{S_0}$ is infinity

We first consider finite $k$ and infinite $S_0$. We look at a couple of simple examples.

$$\operatorname{tr} M = k, \quad \operatorname{tr} M^2 = (\operatorname{tr} M)^2 = k(k+2)$$

$$\operatorname{tr} M^3 = (\operatorname{tr} M)^3 = k(k-1)(k-2) + 3k(k-1)3 + 15k = (k^3 + 6k^2 + 8k)$$

We consider $\operatorname{tr} M^n$ for general $n$. The boundary condition contains $n$ pairs of circles. We label the boundary conditions by a set of non-negative integers $a_1, ..., a_k$ where $a_1 + ... + a_k = n$. The number of flavor i circles is $2a_i$. The number of ways of pairing them up is given by

$$f(a_i) = \frac{1}{a_i!}\binom{2a_i}{2}\binom{2(a_i - 1)}{2}...\binom{2}{2} = \frac{1}{a_i!}\frac{(2a_i)!}{2^{a_i}}$$

So we have

$$\operatorname{tr} M^n = \sum_{a_1+...+a_k=n}\binom{n}{a_1}\binom{n - a_1}{a_2}...\binom{n - a_1 - ... - a_{k-1}}{a_k}f(a_1)...f(a_k) \tag{B.1}$$

$$= \sum_{a_1+...+a_k=n}\frac{n!}{a_1!...a_k!}\frac{(2a_1)!...(2a_k)!}{a_1!...a_k!}\frac{1}{2^{a_1+...+a_k}} \tag{B.2}$$

We consider the generating function of the distribution:

$$\operatorname{tr} e^{tM} = \sum_n \frac{t^n}{n!}\operatorname{tr} M^n = \sum_{a_1,...a_k}\frac{(2a_1)!}{(a_1!)^2}\left(\frac{t}{2}\right)^{a_1}...\frac{(2a_k)!}{(a_k!)^2}\left(\frac{t}{2}\right)^{a_k}$$

$$= \left( \sum_{m=0}^{\infty} \frac{(2m)!}{(m!)^2} \left( \frac{t}{2} \right)^m \right)^k$$

$$= (1 - 2t)^{-\frac{k}{2}} \tag{B.3}$$

(B.3) is the generating functional for chi-squared distribution. The distribution of $\operatorname{tr} M$ is given by

$$p_{\operatorname{tr} M}(x) = \frac{1}{2^{\frac{k}{2}} \Gamma(\frac{k}{2})} x^{\frac{k}{2}-1} e^{-\frac{x}{2}}$$

It is peaked at $k$ as expected.

## B.2  Distribution of $\operatorname{tr} M$ when $e^{S_0}$ is finite

In this appendix we derive the distribution of $\operatorname{tr} M$ for finite $k$ and finite $e^{S_0}$.

Consider $\operatorname{tr} M^n$. We first pair up the flavor indices. As in equation (B.2), this will give a factor $\frac{(2a_1)!...(2a_k)!}{(a_1!...a_k!)^2} \frac{n!}{2^{a_1+...+a_k}}$. Next, we combine these cylinders into higher topology surfaces. Say, at the end there are $b_m$ components with $2m$ boundaries. We have

$$\sum_{m=0}^{n} m b_m = n$$

The number of ways to obtain this partition is given by

$$\binom{n}{b_1} \binom{n - b_1}{2b_2} \cdots \binom{nb_n}{nb_n} \prod_{m=1}^{n} \left[ \binom{mb_m}{m} \binom{m(b_m - 1)}{m} \cdots \binom{m}{m} \frac{1}{b_m!} \right]$$

$$= \frac{n!}{b_1!...(nb_n)!} \prod_{m=1}^{n} \left[ \frac{(mb_m)!}{(m!)^{b_m} b_m!} \right]$$

$$= \frac{n!}{\prod_{m=1}^{n} [(m!)^{b_m} b_m!]}$$

This partition will give a topological factor

$$e^{S_0(2(b_1+...+b_n))-2n} h^{b_1+..+b_n}$$

where

$$h = \sum_{g=0}^{\infty} e^{-2S_0 g} = \frac{1}{1 - e^{-2S_0}}$$

accounts for the effect of handles.

So we have

$$\operatorname{tr} M^n = \left[ \sum_{a_1+\ldots+a_k=n} \frac{(2a_1)!\ldots(2a_k)!}{(a_1!\ldots a_k!)^2} \frac{n!}{2^n} \right] \left[ \sum_{b_1+2b_2+\ldots+nb_n=n} e^{S_0(2(b_1+\ldots+b_n)-2n)} \frac{n!}{\prod_{m=1}^{n}((m!)^{b_m}b_m!)} h^{b_1+\ldots+b_n} \right]$$

(B.4)

Let's solve it in the following way. Equation (B.4) implies that $\operatorname{tr} M$ is the product of two independent random variables:

$$(\operatorname{tr} M)^n = A^n Z^n$$

For $A$, we solved its distribution in appendix B.1.

$$\overline{e^{tA}} = (1-2t)^{-\frac{k}{2}}$$

(B.5)

For $Z$, its generating function is given by

$$
\begin{aligned}
\overline{e^{sZ}} &= \sum_n \frac{\operatorname{tr} Z^n}{n!} s^n \\
&= \sum_{b_1,b_2\ldots} (se^{-2S_0})^{b_1+2b_2+\cdots} h^{b_1+\ldots+b_n} e^{2S_0(b_1+b_2\ldots+)} \frac{1}{(1!)^{b_1}(2!)^{b_2}\ldots b_1!b_2!\ldots} \\
&= \prod_{m=1}^{\infty} \left( \sum_{b_m=0}^{\infty} \left( \frac{hs^m e^{-2(m-1)S_0}}{m!} \right)^{b_m} \frac{1}{b_m!} \right) \\
&= \prod_{m=1}^{\infty} \exp\left( \frac{hs^m e^{-2mS_0} e^{2S_0}}{m!} \right) = \exp\left( he^{2S_0} \sum_{m=1}^{\infty} \frac{(se^{-2S_0})^m}{m!} \right) \\
&= \exp\left( he^{2S_0} \left( e^{se^{-2S_0}} - 1 \right) \right)
\end{aligned}
$$

(B.6)

We see that $Z$ is a Poisson distribution with rate $\lambda = he^{2S_0}$ and takes value at integer multiples of $e^{-2S_0}$.

The object we are interested in is $\overline{e^{tAZ}} = \operatorname{tr}(e^{tM})$. We have

$$
\begin{aligned}
\operatorname{tr}(e^{tM}) = \overline{e^{tAZ}} &= \int dx p_A(x) \overline{e^{txZ}} \\
&= \int dx p_A(x) \exp\left( he^{2S_0} \left( e^{txe^{-2S_0}} - 1 \right) \right) \\
&= \exp(-he^{2S_0}) \int dx p_A(x) \sum_{n=0}^{\infty} \frac{1}{n!} (he^{2S_0})^n e^{ntxe^{-2S_0}} \\
&= \exp(-he^{2S_0}) \sum_{n=0}^{\infty} \frac{1}{n!} (he^{2S_0})^n \overline{e^{nte^{-2S_0}A}}
\end{aligned}
$$

$$= \exp\left(-he^{2S_0}\right)\sum_{n=0}^{\infty}\frac{1}{n!}(he^{2S_0})^n\left(1-2nte^{-2S_0}\right)^{-\frac{k}{2}}, \tag{B.7}$$

where in going from the first to the second line we used (B.6), and to arrive at the last line we used (B.5).

From the generating functional (B.7), the distribution of $\mathrm{tr}\,M$ is given by

$$
\begin{aligned}
p_{\mathrm{tr}\,M}(x) &= e^{-he^{2S_0}}\delta(x) + \sum_{n=1}^{\infty} p_Z(n)p_A\left(\frac{xe^{2S_0}}{n}\right)\frac{e^{2S_0}}{n} \\
&= e^{-he^{2S_0}}\delta(x) + \sum_{n=1}^{\infty} \frac{(he^{2S_0})^n e^{-he^{2S_0}}}{n!}\frac{e^{2S_0}}{kn}\frac{\left(\frac{k}{2}\right)^{\frac{k}{2}}}{\Gamma\left(\frac{k}{2}\right)}\left(\frac{xe^{2S_0}}{kn}\right)^{\frac{k}{2}-1}e^{-\frac{k}{2}\frac{xe^{2S_0}}{kn}}
\end{aligned} \tag{B.8}
$$

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
