# Peer review of "Closed universes in two dimensional gravity"

_SciPost Physics_

## Round 2 · Referee Report · Anonymous (Referee 1) · 2024-5-3

Report

This paper studies closed universes in JT gravity. The authors show that the closed universe Hilbert space is one-dimensional when non-perturbative effects coming from wormholes are included.

The introduction does a good job of motivating the problem and summarizing key results. The technical details are clearly explained.

Some minor comments about the calculations follow:
(1) In eqs. (3.4) and (3.5), the quantities on the left should have an overline indicating an ensemble average. Without this averaging these equations seem inconsistent.
(2) In eqs. (3.6) and (3.7), a non-standard summation convention is implicitly assumed. It would be better to include the explicit sums.

The discussion section addresses some of the conceptual questions raised by the main results. It would be helpful if the authors could include some discussion about whether these results are specific to JT gravity, which is a rather special theory, or they are also expected to hold for higher dimensional gravity theories. Moreover, the main findings seem similar to the Baby Universe Hypothesis of McNamara and Vafa. Any connections of lack thereof should be clarified.

In conclusion, I believe that this paper meets the standards of publication in this journal. My recommendation is that the paper should be published.

Recommendation

Publish (meets expectations and criteria for this Journal)

---

## Round 2 · Referee Report · Anonymous (Referee 3) · 2024-5-23

Strengths

  1. The topic of the work is timely and interesting
  2. The models are simple and tractable, with many analytic results possible
  3. The calculations are well-described and complete

Weaknesses

  1. Referencing and discussion of prior literature in relation to the present work needs improvement
  2. The conceptual setup for the calculations needs improvement

Report

This paper studies two-dimensional closed universes in JT gravity coupled to matter fields as well as in a simpler topological model based on the Marolf-Maxfield model. The introduction presents an overview of the results, section 2 contains various perturbative calculations in JT+matter, section 3 discusses non-perturbative aspects in the topological model, section 4 contains comments and outlook, and there are a few technical appendices.

I think the paper is addressing a timely topic, and the technical computations are correct and well-described. They avoid technical subtleties with 1-loop matter determinants by working with heavy matter in the worldline approximation. I think this is a good technical paper as is, but it needs more discussion of the choice of models, the significance and novelty of the results, and the relation to the literature to meet SciPost’s publication standard.

My opinion is based on the following issues:

First, the references to prior work need serious improvement. The citations mentioned in a single sentence at the end of the outline should be moved to the start of the introduction and properly discussed. As written, the introduction is obscuring a significant body of work in which similar conclusions have been reached. There are other references that have just been omitted, for example, some of the works of McInnes (e.g. https://arxiv.org/abs/hep-th/0403104) which considered cosmologies obtained from Maldacena-Maoz type constructions. Marolf-Maxfield is not cited until Section 3 despite the results of their work being discussed in the introduction. I also noticed that tensor networks are mentioned with no references.

Second, the paper needs a clearer discussion of the major lessons learned and how they relate to prior works. The introduction is framed in terms of the problems of quantum gravity in closed universes and cosmology. As the authors clearly acknowledge, their chosen models are very far from realistic in this context. If the goal is to identify from the calculations some new potential lessons, what are these? One example might be the conclusion that the Hilbert space of a closed universe is one dimensional, but this argument has been discussed in the literature, e.g. reference 10 for dS. The related fact that distinct semi-classical states can have non-zero overlap diagnosed by wormhole geometries is also known in several contexts, including black holes and other holographic models of cosmology. One virtue of this work is the explicit wave functions obtained from the perturbative study of JT+matter. These calculations are concrete and welcome, but, for example, how does the discussion in section 2.2.2 compare to other models of observers in closed universes?

Third, I had some trouble with the basic setup of these calculations, although I found this became clearer as I proceeded through the paper. For example, unless I missed it, the H appearing in tr( exp( - beta H) O_i ) is never defined. I think from context this is a boundary Hamiltonian of some putative holographic dual of JT+matter, but whatever the interpretation, it should be specified. Relatedly, given our knowledge of JT, this is presumably not a single Hamiltonian but one instance of an ensemble. In section 4.3, this is finally discussed, but I think readers not familiar with this literature would really benefit from a more explicit setup in the introduction. This is also important for the argument about the dimensionality of the Hilbert space, since tr( exp(- beta H) O_i ), if taken literally as a dual field theory object, is just an erratic sample-dependent number. If we are calling this object a “state” then it is immediate that, being a number, it can only correspond to a one-dimensional Hilbert space. It would be really helpful to clarify the conceptual setup for readers.

Fourth, in terms of models, I found the jump from JT+matter to the non-perturbative model based on Marolf-Maxfield to be confusing. The paper states that the conclusions of the topological model should also hold in JT+matter, so why not just work entirely with that model? I know that JT+matter is technically more involved, but on the other hand, if one of the central mysteries is how to reconcile the perturbative and non-perturbative descriptions, shouldn’t we study them both in one model?

In summary, I think it is valuable to see these results worked out in simple concrete models, and the calculations are clear with many useful figures. However, in my opinion, many of the conclusions have been discussed elsewhere in the literature (one-dimensional hilbert space, large overlaps from wormholes, closed universe cosmologies with negative cosmological constant), at least in closely related form. Therefore, I think the paper needs to offer greater conceptual clarity and/or further discussion of the new lessons learned to meet SciPost’s standards.

Recommendation

Ask for major revision

---

## Round 2 · Referee Report · Anonymous (Referee 2) · 2024-5-23

Report

This paper is quite nice, it uses a toy version of JT gravity (with negative cosmological constant) coupled to matter to study some of the puzzling issues related to doing quantum cosmology in a closed universe. In particular they are able to do a number of precise calculations related to the idea that the Hilbert space of a closed universe on a global slice should be one dimensional. It should definitely be published!

I have only two minor technical suggestions:

1) I think the wave function in equation 2.6 can be obtained somewhat more simply directly in the covariant phase space formalism, since the wave function of a $\phi_c$ eigenstate in the $b$ basis is $e^{i\phi_c b}$ and more general slices of constant dilaton we have $b=L/\sqrt{1-\frac{\phi_0^2}{\phi_c^2}}$ from the solution 2.3. The square root is in the numerator in 2.6 however, while this argument would put it in the denominator. Perhaps there is a typo?

2) The $\chi^2$ distribution 3.17 can be obtained more directly from the Gaussian integral 3.16 by simply changing to polar coordinates for the vector $A_i$.

Recommendation

Publish (surpasses expectations and criteria for this Journal; among top 10%)

---

## Editorial Decision

resubmitted